# ENHANCING NEURAL SUBSET SELECTION: INTEGRATING BACKGROUND INFORMATION INTO SET REPRESENTATIONS

**Binghui Xie**[1], **Yatao Bian**[2], **Kaiwen zhou**[1], **Yongqiang Chen**[1]

[1]The Chinese University of Hong Kong, [2]Tencent AI Lab, [3]Hong Kong Baptist University
`{bhxie21,kwzhou,yqchen,wei,jcheng}@cse.cuhk.edu.hk`

**Peilin Zhao**[2], **Bo Han**[3], **Wei Meng**[1], **James Cheng**[1]

`yatao.bian@gmail.com,masonzhao@tencent.com,bhanml@comp.hkbu.edu.hk`

## ABSTRACT

Learning neural subset selection tasks, such as compound selection in AI-aided drug discovery, have become increasingly pivotal across diverse applications. The existing methodologies in the field primarily concentrate on constructing models that capture the relationship between utility function values and subsets within their respective supersets. However, these approaches tend to overlook the valuable information contained within the superset when utilizing neural networks to model set functions. In this work, we address this oversight by adopting a probabilistic perspective. Our theoretical findings demonstrate that when the target value is conditioned on both the input set and subset, it is essential to incorporate an *invariant sufficient statistic* of the superset into the subset of interest for effective learning. This ensures that the output value remains invariant to permutations of the subset and its corresponding superset, enabling identification of the specific superset from which the subset originated. Motivated by these insights, we propose a simple yet effective information aggregation module designed to merge the representations of subsets and supersets from a permutation invariance perspective. Comprehensive empirical evaluations across diverse tasks and datasets validate the enhanced efficacy of our approach over conventional methods, underscoring the practicality and potency of our proposed strategies in real-world contexts.

## 1 INTRODUCTION

The prediction of set-valued outputs plays a crucial role in various real-world applications. For instance, anomaly detection involves identifying outliers from a majority of data (Zhang et al., 2020), and compound selection in drug discovery aims to extract the most effective compounds from a given compound database (Gimeno et al., 2019). In these applications, there exists an implicit learning of a set function (Rezatofighi et al., 2017; Zaheer et al., 2017) that quantifies the utility of a given set input, where the highest utility value corresponds to the most desirable set output.

More formally, let's consider the compound selection task: given a compound database $V$, the goal is to select a subset of compounds $S^* \subseteq V$ that exhibit the highest utility. This utility can be modeled by a parameterized utility function $F_\theta(S; V)$, and the optimization criteria can be expressed as:

$$S^* = \arg\max_{S \in 2^V} F_\theta(S; V). \tag{1}$$

One straightforward method is to explicitly model the utility by learning $U = F_\theta(S; V)$ using supervised data in the form of $\{(S_i, V_i), U_i\}_{i=1}^N$, where $U_i$ represents the true utility value of subset $S_i$ given $V_i$. However, this training approach becomes prohibitively expensive due to the need for constructing a large amount of supervision signals (Balcan & Harvey, 2018).

To address this limitation, another way is to solve Eq.1 with an implicit learning approach from a probabilistic perspective. Specifically, it is required to utilize data in the form of $\{(V_i, S_i^*)\}_{i=1}^N$, where $S_i^*$ represents the optimal subset corresponding to $V_i$. The goal is to estimate $\theta$ such that

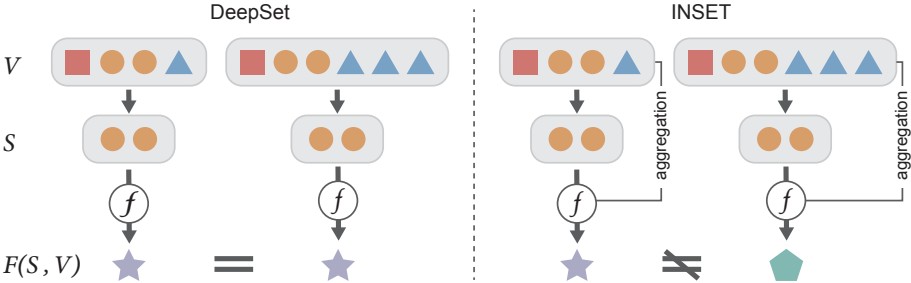

Figure 1: (**Left**): The DeepSet-style models only focus on processing the subset $S$. (**Right**): In contrast, INSET not only identifies the subset $S$ but also takes the identification of $V$ into account, which parameterizes the information that $S$ is a subset of $V$ during the training process.

Eq. 1 holds for all possible $(V_i, S_i)$. During practical training, with limited data $(S_i^*, V_i)_{i=1}^N$ sampled from the underlying data distribution $P(S, V)$, the empirical log likelihood $\sum_{i=1}^N [\log p_\theta(S^*|V)]$ is maximized among all data pairs $\{S, V\}$, where $p_\theta(S|V) \propto F_\theta(S; V)$ for all $S \in 2^V$. To achieve this objective, Ou et al. (2022) proposed to use a variational distribution $q(Y|S, V)$ to approximate the distribution of $P(S|V)$ within the variational inference framework, where $Y \in [0, 1]^{|V|}$ represents a set of $|V|$ independent Bernoulli distributions, representing the odds or probabilities of selecting element $i$ in an output subset $S$. (More details can be found in Appendix D.1.) Thus, the main challenge lies in characterizing the structure of neural networks capable of modeling hierarchical permutation invariant conditional distributions. These distributions should remain unchanged under any permutation of elements in $S$ and $V$ while capturing the interaction between them.

However, the lack of guiding principles for designing a framework to learn the permutation invariant conditional distribution $P(Y|S, V)$ or $F(S, V)$ has been a challenge in the literature. A commonly used approach in the literature involves employing an encoder to generate feature vectors for each element in $V$. These vectors are then fed into DeepSets (Zaheer et al., 2017), using the corresponding supervised subset $S$, to learn the permutation invariant set function $F(S)$. However, this procedure might overlook the interplay between $S$ and $V$, thereby reducing the expressive power of models. See Figure 1 for an illustrative depiction of this concept.

To address these challenges, our research focuses on the aggregation of background information from the superset $V$ into the subset $S$ from a symmetric perspective. Initially, we describe the symmetry group of $(S, V)$ during neural subset selection, as outlined in Section 3.2. Specifically, the subset $S$ is required to fulfill permutation symmetry, while the superset $V$ needs to satisfy a corresponding symmetry group within the nested sets scheme. We denote this hierarchical symmetry of $(S, V)$ as $\mathcal{G}$. Subsequently, we theoretically investigate the connection between functional symmetry and probabilistic symmetry within $F(S, V)$ and $P(Y|S, V)$, indicating that the conditional distribution can be utilized to construct a neural network that processes the *invariant sufficient representation* of $(S, V)$ with respect to $\mathcal{G}$. These representations, defined in Section 3.3, are proven to satisfy *Sufficiency* and *Adequacy*, which means such representations retain the information of the prediction $Y$ while disregarding the order of the elements in $S$ or $V$. Building upon the above theoretical results, we propose an interpretable and powerful model called INSET (Invariant Representation of Subsets) for neural subset selection in Section 3.4. INSET incorporates an information aggregation step between the invariant sufficient representations of $S$ and $V$, as illustrated in Figure 1. This ensures that the model's output can approximate the relationship between $Y$ and $(S, V)$ while being unaffected by the transformations of $\mathcal{G}$. Furthermore, in contrast to previous works that often disregard the information embedded within the set $V$, our exceptional model (INSET) excels in identifying the superset $V$ from which the subset $S$ originates.

In summary, we makes the following contributions. Firstly, we approach neural set selection from a symmetric perspective and establish the connection between functional symmetry and probabilistic symmetry in $P(Y|S, V)$, which enables us to characterize the model structure. Secondly, we introduce INSET, an effective and interpretable approach model for neural subset selection. Lastly, we empirically validate the effectiveness of INSET through comprehensive experiments on diverse datasets, encompassing tasks such as product recommendation and set anomaly detection.

## 2 RELATED WORK

**Encoding Interactions for Set Representations.** Designing network architectures for set-structured input has emerged as a highly popular research topic. Several prominent works, including those (Ravanbakhsh et al., 2017; Edwards & Storkey, 2016; Zaheer et al., 2017; Qi et al., 2017a; Horn et al., 2020; Bloem-Reddy & Teh, 2020) have focused on constructing permutation equivariant models using standard feed-forward neural networks. These models demonstrate the ability to universally approximate continuous permutation-invariant functions through the utilization of set-pooling layers. However, existing approaches solely address the representation learning at the set level and overlook interactions within sub-levels, such as those between elements and subsets.

Motivated by this limitation, subsequent studies have proposed methods to incorporate richer interactions when modeling invariant set functions for different tasks. For instance, (Lee et al., 2019b) introduced the use of self-attention mechanisms to process elements within input sets, naturally capturing pairwise interactions. Murphy et al. (2018) proposed Janossy pooling as a means to encode higher-order interactions within the pooling operation. Further improvements have been proposed by (Kim, 2021; Li et al., 2020), among others. Additionally, Bruno et al. (2021); Willette et al. (2022) developed techniques to ensure Mini-Batch Consistency in set encoding, enabling the provable equivalence between mini-batch encodings and full set encodings by leveraging interactions. These studies emphasize the significance of incorporating interactions between different components.

**Information-Sharing in Neural Networks.** In addition to set learning tasks, the interaction between different components holds significance across various data types and neural networks. Recent years have witnessed the development of several deep neural network-based methods that explore hierarchical structures. For Convolutional Neural Networks (CNNs), various hierarchical modules have been proposed by Deng et al. (2014); Murthy et al. (2016); Xiao et al. (2014); Chen et al. (2020); Ren et al. (2020; 2019) to address different image-related tasks. In the context of graph-based tasks, (Defferrard et al., 2016; Cangea et al., 2018; Gao & Ji, 2019; Ying et al., 2018b; Huang et al., 2019; Ying et al., 2018a; Jin et al., 2020; Han et al., 2022), and others have put forth different methods to learn hierarchical representations. The focus of these works lies in capturing local information effectively and integrating it with global information.

However, the above works ignore the symmetry and expressive power in designing models. Motivated by this, Maron et al. (2020); Wang et al. (2020) proposed how to design linear equivariant and invariant layers for learning hierarchical symmetries to handle per-element symmetries. Moreover, there are some works proposed for different tasks considering symmetry and hierarchical structure, e.g., (Han et al., 2022; Ganea et al., 2022). Our method differs from previous work by focusing on generating a subset $S \in V$ as the final output, rather than output the entire set $V$. Besides, INSET embraces a probabilistic perspective, aligning with the nature of the Optimal Subset (OS) oracle.

## 3 METHOD

### 3.1 BACKGROUND

Let's consider the ground set composed of $n$ elements, denoted as $x_i$, i.e., $V = \{x_1, x_2, ..., x_n\}$. In order to facilitate the proposition of Property 3.1, we describe $V$ as a collection of several disjoint subsets, specifically $V = \{S_1, \ldots, S_m\}$, where $S_i \in \mathbb{R}^{n_i \times d}$. Here, $n_i$ represents the size of subset $S_i$, and each element $x_i \in \mathcal{X}$ is represented by a $d$-dimensional tensor. It is worth noting that, without loss of generality, we can treat $S_i$ as individual elements, i.e., $n_i = 1$. As an example of neural subset selection, the task involves encoding subsets $S_i$ into representative vectors to predict the corresponding function value $Y \in \mathcal{Y}$, as discussed in the introduction section. Existing methods such as (Zaheer et al., 2017) and (Ou et al., 2022) directly select $S_i$ from the encoding embeddings of all elements in $V$, and then input $S_i$ into feed-forward networks. However, these methods approximate the function $F(S_i, V)$ using only the explicit subsets $S_i$, which can be suboptimal since the function also relies on information from the ground set $V$. Furthermore, this approach leads to a conditional distribution $P(Y|S)$ instead of the desired $P(Y|S, V)$. Throughout this study, we assume that all random variables take values in standard Borel spaces, and all introduced maps are measurable.

In this section, we introduce a principled approach for encoding subset representations that leverages background information from the entire input set $V$ to achieve better performance. Additionally,

our theoretical results naturally align with the task of neural subset selection in OS Oracle, as they focus on investigating the probabilistic relationship between $Y$ and $(S, Y)$, which also establishes a connection between the conditional distribution and the functional representation of both $S$ and $V$. By linking the functional representation to the conditional distribution, our results also provide insights into constructing a neural network that effectively approximates the desired function $F(S, V)$.

### 3.2 THE SYMMETRIC INFORMATION FROM SUPERSETS

When considering the invariant representation of $S$ alone, we can directly utilize DeepSets with a max-pooling operation. However, incorporating background information from $V$ into the representations poses the challenge of determining the appropriate inductive bias for the modeling process. One straightforward approach is to assume the existence of two permutation groups that act independently on $V$ and $S$. However, this assumption is impractical since $S$ is a part of $V$. If we transform $S$, the corresponding adjustments should also be made to $V$. From the perspective of the interaction between subsets and supersets, a natural consideration is to view the supersets as a nested set of sets, i.e., $V = S_1 \cup S_2 \cup \cdots \cup S_m$, where $S_i \cap S_j = \emptyset$ if $i \neq j$. In this perspective, the symmetric properties will become more evident.

We assume the presence of an outer permutation $\pi_m \in \mathbb{S}_m$ that maps indices of the subsets to new indices, resulting in a reordering of the subsets within $V$. Furthermore, within each subset $S_i$, there exists a permutation group denoted by $h_i \in \mathcal{H}_i$, which captures the possible rearrangements of elements within that specific subset. Each element of $h_i$ represents a distinct permutation on the elements of $S_i$. The symmetry of nested sets of sets, referred to as $\mathcal{R}$, can be defined as the wreath product of the symmetric group $\mathbb{S}_m$ (representing outer permutations on the $m$ subsets) and the direct product of the permutation groups associated with each subset ($\mathcal{G}_1 \times \mathcal{G}_2 \times \cdots \times \mathcal{G}_m$). Formally, $\mathcal{R} = S_m \wr (\mathcal{G}_1 \times \mathcal{G}_2 \times \cdots \times \mathcal{G}_m)$. Therefore, for any transformation $r \in \mathcal{R}$ acting on $V$, there must exist a corresponding $h \in \mathcal{H}$ acting on $S$. Keeping this in mind, we define the conditional distribution $P(Y|S, V)$ to adhere to the following property:

**Property 3.1.** *Let $S \in \mathcal{S}, V \in \mathcal{V}$ and $Y \in \mathcal{Y}$, where $\mathcal{H}$ and $\mathcal{R}$ act on $\mathcal{S}$ and $\mathcal{V}$, respectively. Then, the conditional distribution $P(Y|S, V)$ of $Y$ give $(S, V)$ is said to be invariant under a group $\mathcal{H}$ and $\mathcal{R}$ if and only if:*

$$P(Y \mid S, V) = P(Y \mid g \cdot (S, V)) = P(Y \mid h \cdot S, r \cdot V) \quad \text{for any } h \in \mathcal{H} \quad \text{and} \quad r \in \mathcal{R} \ .$$

In this context, we denote the composite group $\mathcal{G} = \mathcal{H} \times \mathcal{R}$, which acts on the product space $S \times V$. We have now clarified the specific inductive bias that should be considered when characterizing neural networks. In the subsequent subsection, we will delve into the exploration of constructing neural networks that fulfill this property.

### 3.3 INVARIANT SUFFICIENT REPRESENTATION

Functional and probabilistic notions of symmetries represent two different approaches to achieving the same goal: a principled framework for constructing models from symmetry considerations. To characterize the precise structure of the neural network satisfying Property 3.1, we need to use a technical tool, that transfers a conditional probability distribution $P(Y|S, V)$ into a representation of $Y$ as a function of statics of $(S, V)$ and random noise, i.e., $f(\xi, M(S, V))$. Here, $M$ are maps, which are based on the idea that a statistic may contain all the information that is needed for an inferential procedure. There are mainly two terms as *Sufficiency* and *Adequacy*. The ideas go hand-in-hand with notions of symmetry: while invariance describes information that is irrelevant, sufficiency and adequacy describe the information that is relevant.

There are various methods to describe sufficiency and adequacy, which are equivalent under some constraints. For convenience and completeness, we follow the concept from (Halmos & Savage, 1949; Bloem-Reddy & Teh, 2020). We begin by defining the sufficient statistic as follows, where $\mathcal{B}_{\mathcal{X}}$ represents the Borel $\sigma$-algebra of $\mathcal{X}$:

**Definition 3.2.** *Assume $M : \mathcal{S} \times \mathcal{V} \to \mathcal{M}$ a measurable map and there is a Markov kernel $k : \mathcal{B}_{\mathcal{X}} \times \mathcal{S} \times \mathcal{V} \to \mathbb{R}_+$ such that for all $X \in \mathcal{X}$ and $m \in \mathcal{M}$, $P(\bullet \mid M(S, V) = m) = k(\bullet, m)$. Then $M$ is a **sufficient statistic** for $P(S, V)$.*

This definition characterizes the information pertaining to the distribution of $(S, V)$. More specifically, it signifies that there exists a single Markov kernel that yields the same conditional distribution of $(S, V)$ conditioned on $M(S, V) = m$, regardless of the distribution $P(S, V)$. It is important to note that if $S \nsubseteq V$, the corresponding value of $M(S, V)$ would be zero, which is an invalid case. When examining the distribution of $Y$ conditioned on $S$ and $V$, an additional definition is required:

**Definition 3.3.** *Let $M : \mathcal{S} \times \mathcal{V} \to \mathcal{M}$ be a measurable map and assume $M$ is sufficient for $P(S, V)$. If for all $s \in \mathcal{S}$, $v \in \mathcal{V}$ and $y \in \mathcal{Y}$,*

$$P(Y \in \bullet \mid S = s, V = v) = P(Y \in \bullet \mid M(S, V) = m). \tag{2}$$

*Then, $M$ serves as an **adequate statistic** of $(S, V)$ for $Y$, and also acts as the sufficient statistic.*

Actually, Equation (2) is equivalent to conditional independence of $Y$ and $(S, V)$, given $M(S, V)$, i.e., $Y \perp\!\!\!\perp_{M(X)} X$, This is also called $M$ d-separates $(S, V)$ and $Y$. In other words, if our goal is to approximate the invariant conditional distribution $P(Y|S, V)$, we can first seek an invariant representation of $(S, V)$ under $\mathcal{G}$, which also acts as an adequate statistic for $(S, V)$ with respect to $Y$. Consequently, modeling the relationship between $(S, V)$ and $Y$ directly is equivalent to learning the relationship between $M(S, V)$ and $Y$, which naturally satisfies Property 3.1.

With the given definitions, it becomes evident that we can discover an invariant representation of $(S, V)$ with respect to the symmetric groups $\mathcal{G}$. This representation is referred to as the **Invariant Sufficient Representation**, signifying that an invariant effective representation should eliminate the information influenced by the actions of $\mathcal{G}$, while preserving the remaining information regarding its distribution. This concept is also referred to as *Maximal Invariant* in some previous literature, such as (Kallenberg et al., 2017; Bloem-Reddy & Teh, 2020).

**Definition 3.4.** *(**Invariant Sufficient Representation**) For a group $\mathcal{G}$ of actions on any $(s, v) \in \mathcal{S} \times \mathcal{V}$, we say $M : \mathcal{S} \times \mathcal{V} \to \mathcal{M}$ is a invariant sufficient representation for space $\mathcal{S} \times \mathcal{V}$, if it satisfies: If $M(s_1, v_1) = M(s_2, v_2)$, then $(s_2, v_2) = g \cdot (s_1, v_1)$ for some $g \in \mathcal{G}$; otherwise, there is no such $g$ that satisfies $(s_2, v_2) = g \cdot (s_1, v_1)$.*

Clearly, the invariant sufficient representation $M$ serves as the sufficient statistic for $(S, V)$. Furthermore, if the conditional distribution $P(Y|S, V)$ is invariant to transformations induced by the group $\mathcal{G}$, we can establish that $M(S, V)$ is an adequate statistic for $(S, V)$, as stated in Corollary 3.6. In other words, $M(S, V)$ can be considered to encompass all the relevant information for predicting the label given $(S, V)$ while eliminating the redundant information about $\mathcal{G}$. Hence, we can construct models that learn the relationship between $M(S, V)$ and $Y$, ultimately resulting in an invariant function $Y = f(S, V)$ under the group $\mathcal{G}$. From a probabilistic standpoint, this implies that $P(Y|S, V) = P(Y|M(S, V))$.

### 3.4 CHARACTERIZING THE MODEL STRUCTURE

Hence, by computing the invariant sufficient representations of $(S, V)$, we can construct a $\mathcal{G}$-invariant layer. This idea can give rise to the following theorem:

**Theorem 3.5.** *Consider a measurable group $\mathcal{G}$ acting on $\mathcal{S} \times \mathcal{V}$. Suppose we select an invariant sufficient representation denoted as $M : \mathcal{S} \times \mathcal{V} \to \mathcal{M}$. In this case, $P(Y|S, V)$ satisfies Property 3.1 if and only if there exists a measurable function denoted as $f : [0, 1] \times \mathcal{S} \times \mathcal{V} \to \mathcal{Y}$ such that the following equation holds:*

$$(S, V, Y) =_{\text{a.s.}} \big(S, V, f(\xi, M(S, V))\big) \quad \text{where } \xi \sim \text{Unif}[0, 1] \text{ and } \xi \perp\!\!\!\perp (S, V);. \tag{3}$$

In this context, the variable $\xi$ represents generic noise, which can be disregarded when focusing solely on the model structure rather than the complete training framework (Bloem-Reddy & Teh, 2020; Ou et al., 2022). Consequently, the theorem highlights the necessity of characterizing the neural networks in the form of $f(M(S, V))$. Moreover, Theorem 3.5 implies that the invariant sufficient representation $M(S, V)$ also serves as an adequate statistic. This can be illustrated as follows:

$$P(Y \in \bullet | S = s, V = v) = P(Y \in \bullet | S = s, V = v, M(S, V) = m) = P(Y \in \bullet | M(S, V) = m).$$

To provide additional precision and clarity, we present the following corollary, which demonstrates that $M(S, V)$ is an adequate statistic of $(S, V)$ for $Y$.

**Corollary 3.6.** *Let $\mathcal{G}$ be a compact group acting measurably on standard Borel spaces $S \times V$, and let $\mathcal{M}$ be another Borel space. Then Any invariant sufficient representation $M : \mathcal{S} \times \mathcal{V} \to \mathcal{M}$ under $\mathcal{G}$ is an adequate statistic of $(S, V)$ for $Y$.*

## 3.5 IMPLEMENTATION

In theory, invariant sufficient representations can be computed by selecting a representative element for each orbit under the group $\mathcal{G}$. However, this approach is impractical due to the high dimensions of the input space and the potentially enormous number of orbits. Instead, in practice, a neural network can be employed to *approximate* this process and generate the desired representations (Zaheer et al., 2017; Bloem-Reddy & Teh, 2020), particularly in tasks involving sets or set-like structures.

However, the approach to approximating such a representation for $(S, V)$ under $\mathcal{G}$ remains unclear. To simplify the problem, we can divide the task of finding the invariant sufficient representation of $S$ and $V$ under $H$ and $R$, respectively, as defined in Section 3.2. This concept is guaranteed by the following proposition:

**Proposition 3.7.** *Assuming that $M_s : \mathcal{S} \to \mathcal{S}_1$ and $M_v : \mathcal{V} \to \mathcal{V}_1$ serve as invariant sufficient representations for $S$ and $V$ with respect to $H$ and $R$, respectively, then there exist maps $f : \mathcal{S}_1 \times \mathcal{V}_1 \to \mathcal{M}$ that establish the invariant sufficient representation of $M$.*

Proposition 3.7 specifically states that we can construct the invariant sufficient representations for $S$ and $V$ individually, as they are comparatively easier to construct compared to $M(S, V)$. In the work of (Bloem-Reddy & Teh, 2020), it is demonstrated that for $S$ under $H$, the empirical measure $M_s(S) = \sum_{s_i \in S} \delta(s_i)$ can be chosen as a suitable invariant sufficient representation. Here, $\delta(s_i)$ represents an atom of unit mass located at $s_i$, such as one-hot embeddings. Additionally, leveraging the proposition established by Zaheer et al. (2017), we can employ $\rho \sum_{s \in S} \phi(s)$ to approximate the empirical measure. This approximation offers a practical and effective approach to constructing the invariant sufficient representation.

**Proposition 3.8.** *If $f$ is a valid permutation invariant function on $S$, it can be approximated arbitrarily close in the form of $f(S) = \rho \left( \sum_{s \in S} \phi(s) \right)$, for suitable transformations $\phi$ and $\rho$.*

During the implementation, an encoder $\phi$ is utilized to generate embeddings for each element. For example, when dealing with sets of images, ResNet can be employed as the encoder. On the other hand, $\rho$ can represent various feedforward networks, such as fully connected layers combined with nonlinear activation functions. Similarly, for the symmetric group $\mathcal{R}$ acting on $V$, Maron et al. (2020) has demonstrated that the *universal approximators* of the invariant sufficient representations are $\sum_{S \in V} \sum_{s \in S} \phi(s)$, which is equivalent to $\sum_{x_j \in V} \phi(x_j)$. Hence, for neural subset selection tasks, when considering a specific subset $S \in V$, The neural network construction is outlined as follows:

$$\theta(S, V) = \sigma \left( \theta_1 (\sum_{i=1}^{n_i} \phi(x_i)) + \theta_2 \left( \sum_{i=1}^{n} \phi(x_j) \right) \right), \tag{4}$$

Here, the feed-forward modules $\theta_1$ and $\theta_2$ are accompanied by a non-linear activation layer denoted by $\sigma$. Intuitively, the inherent simplicity of the structure enables us to utilize the DeepSet module to process all elements in $V$ and integrate them with the invariant sufficient representations of $S$. In practice, there are different ways to integrate the representation of $V$ into the representation of $S$, such as concatenation (Qi et al., 2017a) or addition (Maron et al., 2020). Although this idea is straightforward, in the following section, we will demonstrate how this modification significantly enhances the performance of baseline methods. Notably, this idea has been empirically utilized in previous works, such as (Qi et al., 2017a;b). However, we propose it from a probabilistic invariant perspective. A corresponding equivariant framework was also introduced in Wang et al. (2020), which complements our results in the development of deep equivariant neural networks.

## 4 EXPERIMENTS

The proposed methods are assessed across multiple tasks, including product recommendation, set anomaly detection, and compound selection. To ensure robustness, all experiments are repeated five times using different random seeds, and the means and standard deviations of the results are reported. For additional experimental details and settings, we provide comprehensive information in Appendix E.

**Evaluations.** The main goal of the following tasks is to predict the corresponding $S^\star$ given $V$. Therefore, we evaluate the methods using the mean Jaccard coefficient (MJC) metric. Specifically,

Table 1: Product recommendation results on all categories.

| Categories | Random | PGM | DeepSet | Set Transformer | EquiVSet | INSET |
|---|---|---|---|---|---|---|
| Toys | 0.083 | $0.441 \pm 0.004$ | $0.421 \pm 0.005$ | $0.625 \pm 0.020$ | $0.684 \pm 0.004$ | $\mathbf{0.769 \pm 0.005}$ |
| Furniture | 0.065 | $0.175 \pm 0.007$ | $0.168 \pm 0.002$ | $\mathbf{0.176 \pm 0.008}$ | $0.162 \pm 0.020$ | $0.169 \pm 0.050$ |
| Gear | 0.077 | $0.471 \pm 0.004$ | $0.379 \pm 0.005$ | $0.647 \pm 0.006$ | $0.725 \pm 0.011$ | $\mathbf{0.808 \pm 0.012}$ |
| Carseats | 0.066 | $0.230 \pm 0.010$ | $0.212 \pm 0.008$ | $0.220 \pm 0.010$ | $0.223 \pm 0.019$ | $\mathbf{0.231 \pm 0.034}$ |
| Bath | 0.076 | $0.564 \pm 0.008$ | $0.418 \pm 0.007$ | $0.716 \pm 0.005$ | $0.764 \pm 0.020$ | $\mathbf{0.862 \pm 0.005}$ |
| Health | 0.076 | $0.449 \pm 0.002$ | $0.452 \pm 0.001$ | $0.690 \pm 0.010$ | $0.705 \pm 0.009$ | $\mathbf{0.812 \pm 0.005}$ |
| Diaper | 0.084 | $0.580 \pm 0.009$ | $0.451 \pm 0.003$ | $0.789 \pm 0.005$ | $0.828 \pm 0.007$ | $\mathbf{0.880 \pm 0.007}$ |
| Bedding | 0.079 | $0.480 \pm 0.006$ | $0.481 \pm 0.002$ | $0.760 \pm 0.020$ | $0.762 \pm 0.005$ | $\mathbf{0.857 \pm 0.010}$ |
| Safety | 0.065 | $\mathbf{0.250 \pm 0.006}$ | $0.221 \pm 0.004$ | $0.234 \pm 0.009$ | $0.230 \pm 0.030$ | $0.238 \pm 0.015$ |
| Feeding | 0.093 | $0.560 \pm 0.008$ | $0.428 \pm 0.002$ | $0.753 \pm 0.006$ | $0.819 \pm 0.009$ | $\mathbf{0.885 \pm 0.005}$ |
| Apparel | 0.090 | $0.533 \pm 0.005$ | $0.508 \pm 0.004$ | $0.680 \pm 0.020$ | $0.764 \pm 0.005$ | $\mathbf{0.837 \pm 0.003}$ |
| Media | 0.094 | $0.441 \pm 0.009$ | $0.426 \pm 0.004$ | $0.530 \pm 0.020$ | $0.554 \pm 0.005$ | $\mathbf{0.620 \pm 0.023}$ |

for each data sample $(S^\star, V)$ if the model's prediction is $S'$, then the Jaccard coefficient is given as: $JC(S^\star, S') = \frac{|S^\star \cap S'|}{|S^\star \cup S'|}$. Therefore, the MJC is computed by averaging JC metric over all samples in the test set.

**Baselines.** To show our method can achieve better performance on real applications, we compare it with the following methods:

- **Random.** The results are calculated based on random estimates, which provide a measure of how challenging the tasks are.

- **PGM (Tschiatschek et al., 2018).** PGM is a probabilistic greedy model (PGM) solves optimization Problem 1 with a differentiable extension of greedy maximization algorithm. In our paper, we leverage the results of PGD conducted on various datasets as reported in the study by (Ou et al., 2022).

- **DeepSet (Zaheer et al., 2017).** Here, we use DeepSet as a baseline by predicting the probability of which instance should be in $S^\star$, i.e., learn an invariant permutation mapping $2^V \mapsto [0,1]^{|V|}$. It serves as the backbone in EquiVSet to learn set functions, and can also be employed as a baseline.

- **Set Transformer (Lee et al., 2019a).** Set Transformer, compared with DeepSet, goes beyond by incorporating the self-attention mechanism to account for pairwise interactions among elements. This will make models to capture dependencies and relationships between different elements.

- **EquiVSet (Ou et al., 2022).** EquiVSet uses an energy-based model (EBM) to construct the set mass function $P(S|V)$ from a probabilistic perspective, i.e, they mainly focus on learning a distribution $P(S|V)$ monotonically growing with the utility function $F(S, V)$. This requires to learn a conditional distribution $P(Y|S, V)$ as approximation distribution. Actually, their framework is to approximate symmetric $F(S)$ instead of symmetric $F(S, V)$.

## 4.1 PRODUCT RECOMMENDATION

The task requires models to recommend the most interested subset for a customer given 30 products in a category. We use the dataset (Gillenwater et al., 2014a) from the Amazon baby registry for this experiment, which includes many product subsets chosen by various customers. Amazon classifies each item on a baby registry as being under one of several categories, such as "Health" and "Feeding". Moreover, each product is encoded into a 768-dimensional vector by the pre-trained BERT model based on its textual description. Table 1 reports the performance of all the models across different categories. Out of the twelve cases evaluated, INSET performs best in ten of them, except for Furniture and Safety tasks. The discrepancy in performance can be attributed to the fact that our method is built upon the EquiVSet framework, with the main modification being the model structure for modeling $F(S, V)$. Consequently, when EquiVSet performs poorly, it also affects the performance of INSET. Nonetheless, it is worth noting that INSET consistently outperforms EquiVSet and achieves significantly better results than other baselines in the majority of cases. The margin of improvement is substantial, demonstrating the effectiveness and superiority of INSET.

Table 2: Set anomaly detection and compound selection results

| | ANOMALY DETECTION | | | COMPOUND SELECTION | |
|---|---|---|---|---|---|
| | **Double MNIST** | **CelebA** | **F-MNIST** | **PDBBind** | **BindingDB** |
| RANDOM | 0.0816 | 0.2187 | 0.193 | 0.099 | 0.009 |
| PGM | $0.300 \pm 0.010$ | $0.481 \pm 0.006$ | $0.540 \pm 0.020$ | $0.910 \pm 0.010$ | $0.690 \pm 0.020$ |
| DEEPSET | $0.111 \pm 0.003$ | $0.440 \pm 0.006$ | $0.490 \pm 0.020$ | $0.901 \pm 0.011$ | $0.710 \pm 0.020$ |
| SET TRANSFORMER | $0.512 \pm 0.005$ | $0.527 \pm 0.008$ | $0.581 \pm 0.010$ | $0.919 \pm 0.015$ | $0.715 \pm 0.010$ |
| EQUIVSET | $0.575 \pm 0.018$ | $0.549 \pm 0.005$ | $0.645 \pm 0.010$ | $0.924 \pm 0.011$ | $0.721 \pm 0.009$ |
| INSET | $\mathbf{0.707 \pm 0.010}$ | $\mathbf{0.580 \pm 0.012}$ | $\mathbf{0.721 \pm 0.021}$ | $\mathbf{0.935 \pm 0.008}$ | $\mathbf{0.734 \pm 0.010}$ |

## 4.2 SET ANOMALY DETECTION

We conduct set anomaly detection tasks on three real-world datasets: the double MNIST (Sun, 2019), the CelebA (Liu et al., 2015b) and the F-MNIST (Xiao et al., 2017). Each dataset is divided into the training, validation, and test sets with sizes of 10,000, 1,000, and 1,000, respectively. For each dataset, we randomly sample $n \in \{2, 3, 4\}$ images as the OS oracle $S^*$. The setting is followed by (Zaheer et al., 2017; Ou et al., 2022). Let's take CelebA as an example. In this case, the objective is to identify anomalous faces within each set solely through visual observation, without any access to attribute values. The CelebA dataset comprises 202,599 face images, each annotated with 40 boolean attributes. When constructing sets, for every ground set $V$, we randomly choose n images from the dataset to form the OS Oracle $S^*$, ensuring that none of the selected images contain any of the two attributes. Additionally, it is ensured that no individual person's face appears in both the training and test sets. Regarding Table 2, it is evident that our model demonstrates a substantial performance advantage over all the baselines. Specifically, in the case of Double MNIST, our model shows a remarkable improvement of 23% compared to EquiVSet, which itself exhibits the best performance among all the baselines considered. This significant margin of improvement highlights the superior capabilities of our model in tackling the given task.

## 4.3 COMPOUND SELECTION IN AI-AIDED DRUG DISCOVERY

The screening of compounds with diverse biological activities and satisfactory ADME (absorption, distribution, metabolism, and excretion) properties is a crucial stage in drug discovery tasks (Li et al., 2021; Ji et al., 2022; Gimeno et al., 2019). Consequently, virtual screening is often a sequential filtering procedure with numerous necessary filters, such as selecting diverse subsets from the highly active compounds first and then removing compounds that are harmful for ADME. After several filtering stages, we reach the optimal compound subset. However, it is hard for neural networks to learn the full screening process due to a lack of intermediate supervision signals, which can be very expensive or impossible to obtain due to the pharmacy's protection policy. Therefore, the models are supposed to learn this complicated selection process in an end-to-end manner, i.e., models will predict $S^*$ only given the optimal subset supervision signals without knowing the intermediate process. However, this is out of the scope of this paper, since the task is much more complex and requires extra knowledge, and thus we leave it as future work.

To simulate the process, we only apply one filter: high bioactivity to acquire the optimal subset of compound selection following (Ou et al., 2022). We conduct experiments using the following datasets: PDBBind (Liu et al., 2015a) and BindingDB (Liu et al., 2007). Table 2 shows that our method performs better than the baselines and significantly outperform the random guess, especially on the BindingDB dataset. Different from the previous tasks, the performance of these methods is closer to each other. That is

Table 3: Ablation Studies on CelebA with different parameters

| | MJC | Parameters |
|---|---|---|
| **Random** | 0.2187 | - |
| **EquiVSet** | $0.549 \pm 0.005$ | 1782680 |
| **EquiVSet (v1)** | $0.554 \pm 0.007$ | 2045080 |
| **EquiVSet (v2)** | $0.560 \pm 0.005$ | **3421592** |
| **INSET** | $\mathbf{0.580 \pm 0.012}$ | 2162181 |

because the structure of complexes (the elements in a set) can provide much information for this task. Thus, the model could predict the activity value of complexes well without considering the interactions between the optimal subset and the complementary. However, our method can still achieve more satisfactory results than the other methods.

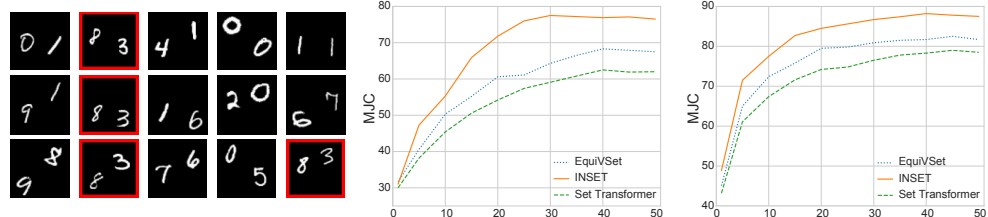

Figure 2: **Left:** A sample from the Double MNIST dataset, comprising $|S^*|$ images displaying the same digit (indicated by the red box). **Right:** The two figures on the right display the validation performance plotted against the number of epochs for Toys and Diaper datasets, respectively. The x-axis represents the epochs.

## 4.4 COMPUTATION COST

The main difference between INSET and EquiVSet is the additional information-sharing module to incorporate the representations of $V$. A possible concern is that the better performance of INSET might come from the extra parameters instead of our framework proposed. To address this concern, we conducted experiments on CelebA datasets. We add an additional convolution layer in the encoders to improve the capacity of EquiVSet. According to the location and size, we propose two variants of EquiVSet, details can be found in the appendix. We report the performance of models with different model sizes in Table 3. It is evident that INSET surpasses all the variants of EquiVSet, clearly demonstrating superior performance. Notably, the improvement achieved through the parameters is considerably less significant when compared to the substantial improvement resulting from the information aggregation process. This highlights the crucial role of information aggregation in driving the overall performance enhancement of INSET.

## 4.5 PERFORMANCE VERSUS TRAINING EPOCHS

In addition to the notable improvement in the final MJC achieved by INSET, we have also observed that incorporating more information from the superset leads to enhanced training speed and better overall performance. To illustrate this, we present two figures depicting the validation performance against the number of training epochs for the Toys and Diaper datasets. It is evident that INSET achieves favorable performance in fewer training epochs. For instance, on the Toy dataset, INSET reaches the best performance of EquiVSet, at approximately epoch 18. Furthermore, around epoch 25, INSET approaches its optimal performance, while EquiVSet and Set Transformer attain their best performance around epoch 40. This highlights the efficiency and effectiveness of INSET in achieving competitive results within a shorter training time.

## 5 CONCLUSION

In this study, we have identified a significant limitation in subset encoding methods, such as neural subset selection, where the output is either the subset itself or a function value associated with the subset. By incorporating the concept of permutation invariance, we reformulate this problem as the modeling of a conditional distribution $P(Y|S, V)$ that adheres to Property 3.1. Our theoretical analysis further reveals that to accomplish this objective, it is essential to construct a neural network based on the invariant sufficient representation of both $S$ and $V$. In response, we introduce INSET, a highly accurate and theoretical-driven approach for neural subset selection, which also consistently outperforms previous methods according to empirical evaluations.

**Limitations and Future Work.** INSET is a simple yet effective method in terms of implementation, indicating that there is still potential for further improvement by integrating additional information, such as pairwise interactions between elements. Furthermore, our theoretical analysis is not limited to set-based tasks; it can be applied to more general scenarios with expanded definitions and theoretical contributions. We acknowledge that these potential enhancements and extensions are left as future work, offering opportunities for further exploration and development.

## 6 ACKNOWLEDGEMENTS

We thanks the reviewers for their valuable comments. Additionally, we extend our thanks to Zijing Ou for his assistance with the code and datasets. This work was supported by CUHK direct grant 4055146. BH was supported by the NSFC General Program No. 62376235, Guangdong Basic and Applied Basic Research Foundation No. 2022A1515011652, Tencent AI Lab Rhino-Bird Gift Fund.

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

## A  APPENDIX

## B  PROOF OF THEOREM 3.5

### B.1  THE CONNECTION BETWEEN FUNCTIONAL AND PROBABILISTIC SYMMETRIES

To prove Theorem 3.5, the main objective is to establish a relationship between the conditional distribution $P(Y|S,V)$ and a functional representation of samples generated from $P(Y|S,V)$ in terms of $(S,V)$ and independent noise $\xi$. This functional representation can be expressed as $Y =_{\text{a.s.}} f(\xi,S,V)$, where $f$ captures the underlying relationship between the variables.

In order to achieve this goal, a valuable technique called the *transducer* (Kallenberg, 2002) or *Noise Outsourcing* (Bloem-Reddy & Teh, 2020) comes into play. This technique allows us to effectively connect the conditional distribution with the functional representation by leveraging the concept of independent noise variables. By applying the transducer or Noise Outsourcing approach, we can establish a clear mapping between the observed variables $(S,V)$, the independent noise $\xi$, and the resulting output $Y$. More formally, we give the corresponding lemma as:

**Lemma B.1** (Conditional independence and randomization). *Let $S,V,Y,Z$ be random elements in some measurable spaces $\mathcal{S}, \mathcal{V}, \mathcal{Y}, \mathcal{Z}$, respectively, where $\mathcal{Y}$ is Borel. Then $Y \perp\!\!\!\perp_Z (S,V)$ if and only if $Y =_{\text{a.s.}} f(\xi,Z)$ for some measurable function $f : [0,1] \times \mathcal{S} \times \mathcal{V} \to \mathcal{Y}$ and some uniform random variable $\xi \perp\!\!\!\perp (S,V,Z)$.*

In the main text, we put forth the idea of utilizing an invariant sufficient representation $M(S,V)$ as an alternative to $(S,V)$ to ensure compliance with symmetry groups. This representation captures the essential information while discarding unnecessary details, making it well-suited for addressing the challenges posed by symmetry.

By employing the invariant sufficient representation $M(S,V)$, we can redefine and refine the aforementioned lemma in a more concise and expressive manner. This approach allows us to establish a direct connection between the conditional distribution $P(Y|S,V)$ and the functional representation $f(\xi,M(S,V))$, where $\xi$ represents independent noise variables. The use of $M(S,V)$ as a replacement for $(S,V)$ enables us to effectively model and analyze the relationship between the input variables and the desired output.

**Lemma B.2.** *Let $S,V$ and $Y$ be random variables with joint distribution $P(S,V,Y)$. Assume there exists a mapping $M : \mathcal{S} \times \mathcal{V} \to \mathcal{M}$, then $M(S,V)$ d-separates $(S,V)$ and $Y$ if and only if there is a measurable function $f : [0,1] \times \mathcal{S} \times \mathcal{V} \to \mathcal{Y}$ such that*

$$(S,V,Y) =_{\text{a.s.}} (S,V,f(\xi,M(S,V))) \quad where \quad \xi \sim \text{Unif}[0,1] \quad and \quad \xi \perp\!\!\!\perp X .$$

*In particular, $Y = f(\xi,M(S,V))$ has distribution $P(Y|S,V)$.*

This lemma implies that if the invariant sufficient representation is capable of d-separating $(S,V)$ and $Y$, it will yield the equation presented in Eq.3 of Theorem 3.5. Subsequently, we must establish why the conditional distribution $P(Y|S,V)$ remains invariant under the group $\mathcal{G}$ if and only if an invariant sufficient representation $M$ exists. This overarching concept can be succinctly summarized in the following lemma:

**Lemma B.3.** *Let $\mathcal{S}, \mathcal{V}$ and $\mathcal{Y}$ be Borel spaces, $\mathcal{G}$ a compact group acting measurably on $(\mathcal{S}, \mathcal{V})$, and $M : \mathcal{S} \times \mathcal{V} \to \mathcal{Y}$ a invariant sufficient representation on $(\mathcal{S}, \mathcal{V})$ under $\mathcal{G}$. If $(S,V)$ is a random element of $(\mathcal{S}, \mathcal{V})$, then its distribution $P(S,V)$ is $\mathcal{G}$-invariant if and only if*

$$P((S,V) \in \bullet \mid M(S,V) = m) = q(\bullet, m) , \tag{5}$$

*for some Markov kernel $q : \mathcal{B}_{\mathcal{S} \times \mathcal{V}} \times \mathcal{M} \to \mathbb{R}_+$. If $P(S,V)$ is $\mathcal{G}$-invariant and $Y$ is any other random variable, then $P(Y|S,V)$ is $\mathcal{G}$-invariant if and only if $Y \perp\!\!\!\perp_{M(S,V)} (S,V)$.*

This lemma serves as a refined version of Lemma 20 presented in (Bloem-Reddy & Teh, 2020). Proving this lemma involves a comprehensive set of definitions and notations, which are beyond the scope of this paper. We encourage interested readers to refer to (Bloem-Reddy & Teh, 2020) for detailed proof.

By leveraging the insights and techniques established above, we are able to establish Theorem 3.5. The theorem provides a formal characterization of the relationship between the invariance of the conditional distribution $P(Y|S,V)$ under the group $\mathcal{G}$ and the existence of an invariant sufficient representation $M(S,V)$.

**Theorem 3.5.** *Consider a measurable group $\mathcal{G}$ acting on $\mathcal{S} \times \mathcal{V}$. Suppose we select an invariant sufficient representation denoted as $M : \mathcal{S} \times \mathcal{V} \to \mathcal{M}$. In this case, $P(Y|S,V)$ satisfies Property 3.1 if and only if there exists a measurable function denoted as $f : [0,1] \times \mathcal{S} \times \mathcal{V} \to \mathcal{Y}$ such that the following equation holds:*

$$(S,V,Y) =_{\text{a.s.}} \big(S,V,f(\xi,M(S,V))\big) \quad \textit{where } \xi \sim \text{Unif}[0,1] \textit{ and } \xi \perp\!\!\!\perp (S,V); . \tag{3}$$

*Proof.* Lemma B.3 plays a crucial role in establishing the conditional independence relationship between $Y$ and $(S,V)$ based on the invariant sufficient representation $M(S,V)$. This lemma demonstrates that when $M(S,V)$ is employed, the variables $Y$ and $(S,V)$ become conditionally independent, meaning that knowledge of $M(S,V)$ is sufficient to explain the relationship between $Y$ and $(S,V)$.

By leveraging the insights provided by Lemma B.3, we can derive the conclusion of Theorem 3.5 with the support of lemma B.2. Lemma B.2 further strengthens the link between the conditional independence relationship and the existence of an invariant sufficient representation. It establishes the notion that the invariance of the conditional distribution $P(Y|S,V)$ under the group $\mathcal{G}$ is directly related to the presence of an invariant sufficient representation $M(S,V)$. □

Given the established conditional independence relationship $Y \perp\!\!\!\perp_{M(S,V)} (S,V)$ as demonstrated in Lemma B.3, we can now proceed to prove the following corollary by examining the definition of adequacy:

**Corollary 3.6.** *Let $\mathcal{G}$ be a compact group acting measurably on standard Borel spaces $S \times V$, and let $\mathcal{M}$ be another Borel space. Then Any invariant sufficient representation $M : \mathcal{S} \times \mathcal{V} \to \mathcal{M}$ under $\mathcal{G}$ is an adequate statistic of $(S,V)$ for $Y$.*

This corollary follows directly from the nature of the conditional independence relationship and the definition of adequacy. The fact that $Y$ and $(S,V)$ are conditionally independent given $M(S,V)$ indicates that the representation $M(S,V)$ contains all the necessary information to explain the relationship between $Y$ and $(S,V)$. In other words, the representation $M(S,V)$ adequately captures the relevant features and factors that influence the conditional distribution of $Y$ given $(S,V)$.

## B.2 SOME REMARKS

Throughout the course of our proof, it is possible that some points may cause confusion. To address this, we present a set of remarks and additional propositions that aim to provide further clarity and insights. Specifically, we address the question of why Eq 3 can lead to an invariant conditional distribution, considering its nature as a joint distribution scheme.

In the probabilistic literature, it is often more convenient to establish the invariance of joint distributions as a starting point. In our case, we focus on the joint distribution $P(S,V,Y)$, which can be considered invariant under certain conditions. This joint distribution is invariant if and only if:

$$(S,V,Y) \overset{\text{d}}{=} (g \cdot (S,V), Y) \quad \text{for all } g \in \mathcal{G} .$$

These conditions ensure that the joint distribution $P(S,V,Y)$ possesses the desired invariance properties required for the subsequent analysis. By establishing an invariant joint distribution, we pave the way for investigating the properties of the conditional distribution $P(Y|S,V)$.

By leveraging the invariance of the joint distribution, we can derive the invariance of the conditional distribution $P(Y|S,V)$. This arises from the fact that the joint distribution scheme inherently captures the relationship between $Y$ and $(S,V)$, allowing us to analyze their conditional distribution in an invariant manner (Kallenberg et al., 2017; Bloem-Reddy & Teh, 2020):

**Proposition B.4.** *Assume a group $\mathcal{G}$ acting on $(\mathcal{S},\mathcal{V})$, and then $P(Y|S,V)$ is $\mathcal{G}$-invariant if and only if $(S,V,Y) \overset{\text{d}}{=} (g \cdot (S,V), Y)$ for all $g \in \mathcal{G}$.*

This proposition establishes a direct correspondence between the invariance of the conditional distribution $P(Y|S,V)$ and the symmetry of the joint distribution $(S,V,Y)$ under the group action. Specifically, it states that the conditional distribution remains invariant if and only if the joint distribution exhibits the same structural patterns and properties when transformed by any element of the group $\mathcal{G}$.

To address another potential source of confusion, we delve into the distinction between $f(\xi, S, V)$ and $f(S,V)$, where the former represents a stochastic function and the latter a deterministic functional model. It is worth noting that deterministic functional models can be viewed as a special case of stochastic functions. In the context of an invariant stochastic function $Y = f(\xi, S, V)$, we can establish the following relationship:

$$\mathbb{E}[Y \mid S, V] = \int_{[0,1]} f(\xi, S, V) \, d\xi = h(S, V) \ , \tag{6}$$

Here, $\mathbb{E}[Y \mid S, V]$ denotes the conditional expectation of $Y$ given $S$ and $V$. By integrating the stochastic function $f(\xi, S, V)$ with respect to $\xi$ over the range $[0, 1]$, we arrive at the invariant deterministic function $h(S, V)$. Equation 6 establishes a crucial connection between the invariant stochastic function $f(\xi, S, V)$ and the corresponding invariant deterministic function $h(S, V)$. This relationship highlights the interplay between stochasticity and determinism in modeling invariant behavior.

## C    PROOF OF PROPOSITION 3.7

**Definition 3.4.** *(Invariant Sufficient Representation)* *For a group $\mathcal{G}$ of actions on any $(s, v) \in \mathcal{S} \times \mathcal{V}$, we say $M : \mathcal{S} \times \mathcal{V} \to \mathcal{M}$ is a invariant sufficient representation for space $\mathcal{S} \times \mathcal{V}$, if it satisfies: If $M(s_1, v_1) = M(s_2, v_2)$, then $(s_2, v_2) = g \cdot (s_1, v_1)$ for some $g \in \mathcal{G}$; otherwise, there is no such $g$ that satisfies $(s_2, v_2) = g \cdot (s_1, v_1)$.*

It is important to note that the definition of the invariant sufficient representation is formulated with respect to the variables $(S, V)$. To facilitate comprehension, let us revisit the definition of the invariant sufficient representation. Furthermore, to enhance clarity and ease of understanding, we will also provide the corresponding invariant sufficient representation for the marginal distribution $X$, which can represent either $S$ or $V$.

**Definition C.1.** *For a group $\mathcal{G}$ of actions on any $x \in \mathcal{X}$, we say $M : \mathcal{X} \to \mathcal{M}$ is a invariant sufficient representation for some space $\mathcal{X}$, if it satisfies: If $M(x_1) = M(x_2)$, then $x_2 = g \cdot x_1$ for some $g \in \mathcal{G}$; otherwise, there is no such $g$ that satisfies $x_2 = g \cdot x_1$.*

**Proposition 3.7.** *Assuming that $M_s : \mathcal{S} \to \mathcal{S}_1$ and $M_v : \mathcal{V} \to \mathcal{V}_1$ serve as invariant sufficient representations for $S$ and $V$ with respect to $H$ and $R$, respectively, then there exist maps $f : \mathcal{S}_1 \times \mathcal{V}_1 \to \mathcal{M}$ that establish the invariant sufficient representation of $M$.*

*Proof.* For any $(s, v)$ Consider the function $f(M_s(s), M_v(v))$, where $M_s$ and $M_v$ represent the mappings from $S$ and $V$ to their corresponding invariant sufficient representations. We aim to show that $f(M_s(S), M_v(V))$ serves as the invariant sufficient representation of $(S, V)$ under the group $\mathcal{G}$.

First, let's examine the three possible cases. If $s_1 \neq s_2$ while $v_1 = v_2$, it is evident that if $f(M_s(s_1), M_v(v_2)) = f(M_s(s_2), M_v(v_2))$, there must exist $g = (h, e) \in \mathcal{G}$ such that $(s_2, v_2) = g \cdot (s_1, v_1)$, where $e$ denotes the identity transformation. Similarly, if $s_1 = s_2$ but $v_1 \neq v_2$, the same argument holds.

Now, let's consider the case where $s_1 \neq s_2$ and $v_1 \neq v_2$. Since $f$ is an injective function, if $f(M_s(s_1), M_v(v_2)) = f(M_s(s_2), M_v(v_2))$, it implies that $M_s(s_1) = M_s(s_2)$ and $M_v(v_1) = M_v(v_2)$. By the definition of the invariant sufficient representation, we can conclude that if $f(M_s(s_1), M_v(v_1)) = f(M_s(s_2), M_v(v_2))$, then $(s_2, v_2) = g \cdot (s_1, v_1)$ for some $g \in \mathcal{G}$. Conversely, if no such $g$ exists to satisfy $(s_2, v_2) = g \cdot (s_1, v_1)$, it implies that $f(M_s(s_1), M_v(v_1)) \neq f(M_s(s_2), M_v(v_2))$.

Therefore, we can conclude that $f(M_s(S), M_v(V))$ serves as the invariant sufficient representation of $(S, V)$ under the group $\mathcal{G}$. This function captures the essential information required to explain the relationship between $(S, V)$ and $Y$, ensuring that the conditional distribution $P(Y|S,V)$ remains invariant under the group action. $\square$

Table 4: The statistics of Amazon product dataset, which is from (Ou et al., 2022)

| Categories | $|\mathcal{D}|$ | $|V|$ | $\sum|S^*|$ | $\mathbb{E}[|S^*|]$ | $\min_{S^*}|S^*|$ | $\max_{S^*}|S^*|$ |
|---|---|---|---|---|---|---|
| Toys | 2,421 | 30 | 9,924 | 4.09 | 3 | 14 |
| Furniture | 280 | 30 | 892 | 3.18 | 3 | 6 |
| Gear | 4,277 | 30 | 16,288 | 3.80 | 3 | 10 |
| Carseats | 483 | 30 | 1,576 | 3.26 | 3 | 6 |
| Bath | 3,195 | 30 | 12,147 | 3.80 | 3 | 11 |
| Health | 2,995 | 30 | 11,053 | 3.69 | 3 | 9 |
| Diaper | 6,108 | 30 | 25,333 | 4.14 | 3 | 15 |
| Bedding | 4,524 | 30 | 17,509 | 3.87 | 3 | 12 |
| Safety | 267 | 30 | 846 | 3.16 | 3 | 5 |
| Feeding | 8,202 | 30 | 37,901 | 4.62 | 3 | 23 |
| Apparel | 4,675 | 30 | 21,176 | 4.52 | 3 | 21 |
| Media | 1,485 | 30 | 6,723 | 4.52 | 3 | 19 |

# D  DETAILS OF NEURAL SUBSET SELECTION IN OS ORACLE

## D.1  THE OBJECTIVE OF NEURAL SUBSET SELECTION IN OPTIMAL SUBSET ORACLE

Our formulation of the optimization objective is based on the framework established in (Ou et al., 2022). Specifically, the optimization objective is to address Equation 1 by adopting an implicit learning strategy grounded in probabilistic reasoning. This approach can be succinctly formulated as follows:

$$\arg\max_\theta \ \mathbb{E}_{\mathbb{P}(V,S)}[\log p_\theta(S^*|V)]$$

$$\text{s.t. } p_\theta(S|V) \propto F_\theta(S;V), \forall S \in 2^V,$$

The important step in addressing this problem involves constructing an appropriate set mass function $p_\theta(S|V)$ that is monotonically increasing in relation to the utility function $F_\theta(S;V)$. To achieve this, we can employ the Energy-Based Model (EBM):

$$p_\theta(S|V) = \frac{\exp(F_\theta(S;V))}{Z}, \ Z := \sum_{S'\subseteq V} \exp(F_\theta(S';V)),$$

In practice, we approximate the EBM by solving a variational approximation

$$\phi^* = argmin_\phi D(q_\phi(Y|S,V))||p_\theta(S|V)),$$

The expression of $q(Y|S,V)$ is defined as follows:

$$q(Y|S,V) = \prod_{i\in S} Y_i \prod_{i\notin S}(1-Y_i), Y \in [0,1]^{|V|}.$$

Next, we would like to explain why $q(Y|S,V)$ can approximate $P(S|V)$. We have defined that $Y \in [0,1]^{|V|}$ and $S = \{0,1\}^{|V|}$. In this case, $Y$ can be viewed as a stochastic version of $S$ since $Y$ can also be generated as $Y = \{0,1\}^{|V|}$ while still satisfying the constraint $Y \in [0,1]^{|V|}$.

To facilitate comprehension, let us consider an illustrative scenario. Suppose we have a ground set $V = \{x_1, x_2, x_3\}$, and the optimal subset $S^*$ is $\{x_1, x_2\}$, which can be represented as $[1,1,0]$. Specifically, we define $P(S^*|V) = 1$, indicating that $S^*$ is the correct subset, while for any $S \neq S^*$, we have $P(S|V) = 0$.

Now, let's examine the case when $Y = [1,1,0]$. In this situation, we can calculate that $q(Y|S^*,V) = 1$. This implies that $q(Y|S^*,V)$ accurately represents the probability of observing $S^*$ given $V$, and it correctly assigns a high probability to the optimal subset.

Moreover, to generate $Y$, we construct an EquiNet (Ou et al., 2022), denoted as $Y = \text{EquiNet}(V;\phi) : 2^V \to [0,1]^{|V|}$. This network takes the ground set $V$ as input and outputs probabilities indicating the likelihood of each element $x \in V$ being part of the optimal subset $S^*$. In the inference stage, EquiNet is employed to predict the optimal subset for a given ground set $V$, using a TopN rounding approach.

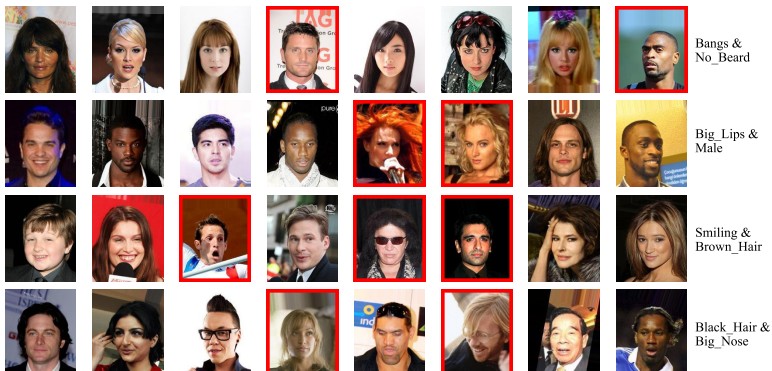

Figure 3: Here is an illustration of the CelebA dataset from the work by (Ou et al., 2022). Each row in the dataset represents a sample, containing a combination of $|S^*|$ anomaly images (highlighted in red boxes) and $8 - |S^*|$ normal images. Notably, within each sample, normal images possess two specific attributes, which are indicated in the rightmost column. In contrast, anomalies lack both of these attributes. This clear distinction between normal images and anomalies allows for a comprehensive analysis and understanding of the dataset's characteristics.

# E  EXPERIMENTAL DETAILS

## E.1  DETAILED DESCRIPTION OF DATASETS

**Amazon Baby Registry Dataset.** The Amazon baby registry data (Gillenwater et al., 2014b) is collected from Amazon with several datasets in different categories, such as toys, furniture, etc. For each category, we are provided with $|V|$ sets of products selected by different customers. We then construct a sample $(S^*, V)$ as follows. We first removed any subset whose optimal subset size $|S^*|$ is greater than or equal to 30. Then we divided the remaining subsets into the training, validation, and test folds with a $1 : 1 : 1$ ratio. Finally, we randomly sampled additional $30 - |S^*|$ products from the same category to construct $(S^\star, V)$. In this way, we constructed a data point $(S^\star, V)$. For completeness, We provide the statistics of the categories in Table. 4 from (Ou et al., 2022).

**Double MNIST.** This dataset includes 1000 photos, ranging from 00 to 99. To construct $(S^*, V)$, we first sampled $|S^*| \in \{2, \dots, 5\}$ images with the same digit as $S^*$. Then we selected $20 - |S^*|$ images with different digits to construct the set $V \backslash S^*$.

**CelebA.** The CelebA dataset contains $202,599$ images and $40$ attributes. We randomly chose two attributes to construct each set $V$ with the size of 8. Then, for each set, we selected $|S^\star| \in \{2, 3\}$ images without the two attributes as the $S^\star$. To facilitate comprehension, we have included an illustrative example of the dataset in Fig. 3, sourced from the work of (Ou et al., 2022).

**PDBBind.** This dataset provides a comprehensive collection of experimentally measured binding affinity data for biomolecular complexes. We used the "refined" part of the whole PDBBind to construct our dataset of subsets, which contain 179 complexes. To construct a data point $(V, S^\star)$, we randomly sampled 30 complexes as the ground set $V$, and then $S^\star *$ was generated by the five most active complexes in $V$. We constructed 1000, 100, and 100 data points for the training, validation, and test split, respectively.

**BindingDB.** BindingDB is a public, web-accessible database of measured binding affinities consisting of $52,273$ drug-targets with small, drug-like molecules. Same as PDBBind, We randomly sampled 300 drug-targets from the BindingDB database to construct the ground set $V$ and select 15 most active drug-target pairs as $S^\star$. Finally, we also generated the training, validation, and test set with the size of 1000, 100, and 100, respectively.

### E.2   THE ARCHITECTURE OF `INSET`

The structure of `INSET` is similar to EquiVSet, while `INSET` has an additional information-sharing component, i.e., EquiVSet uses only one DeepSets layer in the set function modelue, while `INSET` uses two. For completeness, we also provide the detailed architectures of `INSET` in this subsection. Firstly, the structure of DeepSets in `INSET` is as follows:

Table 5: Detailed architectures of the DeepSets.

| Set Function |
| --- |
| $\text{InitLayer}(S, h)$ |
| SumPooling |
| $\text{FC}(h, h_d, \text{ReLU})$ |
| $\text{FC}(h_d, h_d, \text{ReLU})$ |
| $\text{FC}(h, 1, -)$ |

Specifically, $\text{InitLayer}(S, d)$ encodes the set objects into vector representations. $\text{FC}(d, h, f)$ is a fully-connected layer with activation function $f$. In particular, we set $h$ as 256 and $h_d$ as 500, same as (Ou et al., 2022).

In all experiments, the structure of InitLayer will change based on the type of datasets.

**Synthetic datasets.** The synthetic datasets consist of the Tow-Moons and Gaussian-Mixture datasets. Each instance of the set is a two-dimensional vector, which represents the corresponding Cartesian coordinates. In this dataset, the $\text{InitLayer}$ is a one-layer feed-forward neural network $\text{FC}(2, 256, -)$.

**Amazon Baby Registry.** In this datasets, each product is encoded into a 768-dimensional vector by the pre-trained BERT model based on its textual description. Therefore, each element of the set is a 768-dimensional feature vector, and $\text{FC}(768, 256, -)$ will be the $\text{InitLayer}$ to process each embedding of the product.

**Double MNIST.** The double MNIST dataset consists of different digit images with a shape of $(64, 64)$ and we transformed it as $(4096, )$. Then, the $\text{InitLayer}$ is also a fully connected layer as $\text{FC}(4096, 256, -)$.

**CelebA.** The CelebA dataset includes face images in the shape of $(3, 64, 64)$. We used 3-depth convolutional neural networks as the $\text{InitLayer}$. Specifically,

$$\text{ModuleList}([\text{Conv}(32, 3, 2, \text{ReLU}), \text{Conv}(64, 4, 2, \text{ReLU}),$$
$$\text{Conv}(128, 5, 2, \text{ReLU}), \text{MaxPooling}, \text{FC}(128, 256, -)]),$$

where $\text{Conv}(d, k, s, f)$ is a convolutional layer with $d$ output channels, $k$ kernel size, $s$ stride size, and activation function $f$.

**PDBBind.** The PDBBind database consists of experimentally measured binding affinities for biomolecular complexes (Liu et al., 2015a). The atomic convolutional network (ACNN) (Gomes et al., 2017) provides meaningful feature vectors for complexes by constructing nearest neighbor graphs based on the 3D coordinates of atoms and predicting binding free energies. In this work, we used ACNN as the pre-train model and used the output of the second to the lastlayer of the ACNN model to obtain the representations of complexes. Specifically, the $\text{InitLayer}$ is defined as

$$\text{ModuleList}([\text{ACNN}[: -1], \text{FC}(1922, 2048, \text{ReLU}), \text{FC}(2048, 256, -)]),$$

where $\text{ACNN}[: -1]$ denotes the ACNN module without the last prediction layer, whose output dimensionality is 1922.

**BindingDB.** We employ the DeepDTA model (Öztürk et al., 2018) as the based-encoder to transform drug-target pairs as vector representations. The detailed architecture of $\text{InitLayer}$ used in our code is defined as follows:

### E.3   TRAINING DETAILS

We applied the early stopping strategy to train both the baselines and our models as in EquiVSet. Specifically, if the best validation performance is not improved in the continuous 6 epochs, we will

Table 6: Detailed architectures of InitLayer in the BindingDB dataset.

| Drug | Target |
|---|---|
| $\mathrm{Conv}(32, 4, 1, \mathrm{ReLU})$ | $\mathrm{Conv}(32, 4, 1, \mathrm{ReLU})$ |
| $\mathrm{Conv}(64, 6, 1, \mathrm{ReLU})$ | $\mathrm{Conv}(64, 8, 1, \mathrm{ReLU})$ |
| $\mathrm{Conv}(96, 8, 1, \mathrm{ReLU})$ | $\mathrm{Conv}(96, 12, 1, \mathrm{ReLU})$ |
| MaxPooling | MaxPooling |
| $\mathrm{FC}(96, 256, \mathrm{ReLU})$ | $\mathrm{FC}(96, 256, \mathrm{ReLU})$ |
| Concat | |
| $\mathrm{FC}(512, 256, -)$ | |

Table 7: Results in the MJC metric on Two-Moons and Gaussian-Mixture datasets. we leverage the results of baselines as reported in the study by (Ou et al., 2022).

| Method | Two Moons | Gaussian Mixture |
|---|---|---|
| Random | 0.055 | 0.055 |
| PGM | $0.360 \pm 0.020$ | $0.438 \pm 0.009$ |
| DeepSet | $0.472 \pm 0.003$ | $0.446 \pm 0.002$ |
| Set Transformer | $0.574 \pm 0.002$ | $0.905 \pm 0.002$ |
| EquiVSet | $0.584 \pm 0.003$ | $0.908 \pm 0.002$ |
| INSET | $\mathbf{0.590 \pm 0.003}$ | $\mathbf{0.909 \pm 0.002}$ |

stop the training process. The maximum of epochs is set as 100 for each dataset. We saved the models with the best validation performance and evaluated them on the test set. We repeated all experiments 5 times with different random seeds, and the average performance metrics and their standard deviations are reported as the final performances.

The proposed models are trained using the Adam optimizer (Kingma & Ba, 2014) with a fixed learning rate of $1e-4$ and a weight decay rate of $1e-5$. To accommodate the varying model sizes across different datasets, we select the batch size from the set $\{4, 8, 16, 32, 64, 128\}$. Notably, we choose the largest batch size that allows the model to be trained on a single GeForce RTX 2080 Ti GPU, ensuring efficient training.

# F  ADDITIONAL EXPERIMENTS

## F.1  SYNTHETIC EXPERIMENTS

We substantiate the effectiveness of our models by conducting experiments on learning set functions using two synthetic datasets: the two-moons dataset with additional noise of variance $\sigma^2 = 0.1$, and a mixture of Gaussians represented by $\frac{1}{2}\mathcal{N}(\mu_0, \Sigma) + \frac{1}{2}\mathcal{N}(\mu_1, \Sigma)$.

For the Gaussian mixture dataset, we specify the following data generation procedure: i) We first select an index, denoted as $b$, using a Bernoulli distribution with a probability of $\frac{1}{2}$. ii) Next, we sample 10 points from the Gaussian distribution $\mathcal{N}(\mu_b, \Sigma)$ to construct the set $S^*$. iii) Subsequently, we sample 90 points for $V \setminus S^*$ from the Gaussian distribution $\mathcal{N}(\mu_{1-b}, \Sigma)$. We repeat this process to obtain a total of 1,000 samples, which are then divided into training, validation, and test sets.

Both the two-moon dataset and the Gaussian mixture dataset serve as valuable benchmarks for evaluating the performance of our models. By conducting experiments on these datasets and collecting the necessary data points, we are able to demonstrate the efficacy of our approach in learning complex set functions. The results are reported in Table 7.

## F.2  COMPUTATION COST

One of the key distinctions between INSET and EquiVSet lies in the inclusion of an information-sharing module, specifically a DeepSets Layer, in our architecture. However, a legitimate concern

that arises is whether the improved performance of INSET can be solely attributed to the additional parameters introduced by this module, rather than the underlying framework itself. To address this concern and gain deeper insights, we conducted experiments using the CelebA dataset.

In order to enhance the capacity of EquiVSet and enable a fair comparison, we introduced an additional convolution layer within the InitLayer. By doing so, we ensured that both EquiVSet and INSET had comparable model sizes. The experimental results, including the performance of models with different model sizes, are reported in Table 9.

Moreover, we further investigated and analyzed the specific architecture of the initial layer for EquiVSet in two different versions, denoted as EquiVSet(v1) and EquiVSet(v2). For EquiVSet(v1), the initial layer is structured as follows:

$$\text{ModuleList}([\text{Conv}(32, 3, 2, \text{ReLU}), \text{Conv}(64, 4, 2, \text{ReLU}), \text{Conv}(64, 4, 2, \text{ReLU}),$$
$$\text{Conv}(128, 5, 2, \text{ReLU}), \text{MaxPooling}, \text{FC}(128, 256, -)]),$$

The initial layer for EquiVSet(v2) is:

$$\text{ModuleList}([\text{Conv}(32, 3, 2, \text{ReLU}), \text{Conv}(64, 4, 2, \text{ReLU}), \text{Conv}(128, 5, 2, \text{ReLU}),$$
$$\text{Conv}(128, 5, 2, \text{ReLU}), \text{MaxPooling}, \text{FC}(128, 256, -)]),$$

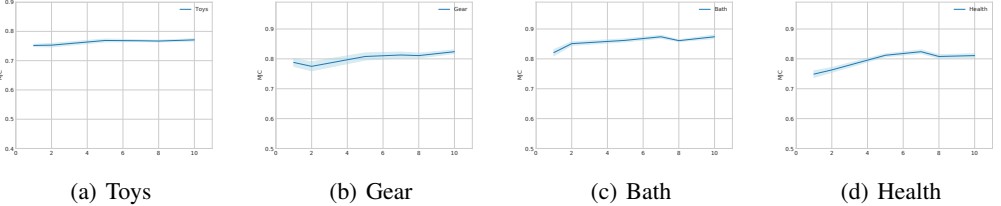

| (a) Toys | (b) Gear | (c) Bath | (d) Health |

Figure 4: Sensitivity analysis of INSET performance under varying numbers of Monte Carlo (MC) sampling.

### F.3 COMPOUND SELECTION

In the main text, we focused on the application of a single filter. However, to provide a more practical perspective, we extended our analysis by simulating the OS (Objective Selector) oracle of compound selection using *two filters*: the high bioactivity filter and the diversity filter. By incorporating these additional filters, we aimed to evaluate the performance of our approach in a more realistic scenario.

The results of this extended analysis are presented in Table 8. These findings shed light on the effectiveness and applicability of our approach when considering multiple filters for compound selection. By incorporating both high bioactivity and diversity filters, we demonstrate the potential of our method to enhance the selection process and improve the overall quality and diversity of the selected compounds.

Table 8: Compound selection results.

| Method | PDBBind | BindingDB |
|---|---|---|
| Random | 0.073 | 0.027 |
| PGM | $0.350 \pm 0.009$ | $0.176 \pm 0.006$ |
| DeepSet | $0.323 \pm 0.004$ | $0.165 \pm 0.005$ |
| Set Transformer | $0.355 \pm 0.010$ | $0.183 \pm 0.004$ |
| EquiVSet | $0.357 \pm 0.005$ | $0.188 \pm 0.006$ |
| INSET | $\mathbf{0.371 \pm 0.010}$ | $\mathbf{0.198 \pm 0.005}$ |

Table 9: Compared with EquiVSet with more parameters

|  | MJC | Parameters |
|---|---|---|
| **Random** | 0.2187 | - |
| **DeepSet** | 0.440±0.006 | 651181 |
| **Set-Transformer** | 0.527±0.008 | 1288686 |
| **EquiVSet** | 0.549±0.005 | 1782680 |
| **EquiVSet (v1)** | 0.554±0.007 | 2045080 |
| **EquiVSet (v2)** | 0.560±0.005 | **3421592** |
| **INSET** | **0.580±0.012** | 2162181 |

## F.4 ABLATION STUDIES

To further verify the robustness of INSET, we have now conducted ablation studies focusing on the Monte-Carlo (MC) sample numbers for each input pair $\{(V_i, S_i^*)\}$. In the context of neural subset selection tasks, our primary aim is to train the model $\theta$ to predict the optimal subset $S^*$ from a given ground set $V$. During training, we sample $m$ subsets from $V$ to optimize our model parameters $\theta$, thereby maximizing the conditional probability distribution $p_\theta(S^*|V)$ among of all pairs of $(S, V)$ for for a given V. In our main experiments, we adhere to EquiVSet's protocol by setting the sample number $m$ to 5 across all the tasks. The empirical results depicted in Figure 4 demonstrate that INSET consistently achieves satisfactory results, even with decreasing values of $m$.

## F.5 EXPERIMENTS ON SET ANOMALY DETECTION.

In this experiment, we further perform set anomaly detection on CIFAR-10. Following the setup of (Ou et al., 2022), we randomly sample $n \in \{2, 3\}$ images as the OS oracle $S^*$, and then select $8 - |S^*|$ images with different labels to construct the set $V \backslash S^*$. We finally obtain the training, validation, and test set with the size of $10,000, 1,000, 1,000$, respectively. We report all the set anomaly detection results in Table 10. It is obviously that INSET outperform the baselines significantly across different datasets on set anomaly detection tasks.

Table 10: Empirical results of set anomaly detection Tasks

|  | Double MNIST | CelebA | F-MNIST | CIFAR-10 |
|---|---|---|---|---|
| RANDOM | 0.0816 | 0.2187 | 0.1930 | 0.1926 |
| PGM | 0.300 ± 0.010 | 0.481 ± 0.006 | 0.540 ± 0.020 | 0.450 ± 0.020 |
| DEEPSET | 0.111 ± 0.003 | 0.440 ± 0.006 | 0.490 ± 0.020 | 0.320 ± 0.008 |
| SET TRANSFORMER | 0.512 ± 0.005 | 0.527 ± 0.008 | 0.581 ± 0.010 | 0.650 ± 0.023 |
| EQUIVSET | 0.575 ± 0.018 | 0.549 ± 0.005 | 0.645 ± 0.010 | 0.630 ± 0.012 |
| INSET | **0.707 ± 0.010** | **0.580 ± 0.012** | **0.710 ± 0.021** | **0.712 ± 0.020** |

