# OpenReview forum: "Enhancing Neural Subset Selection: Integrating Background Information into Set Representations"
_ICLR.cc/2024/Conference — ICLR 2024 poster_

### Official Review · Reviewer_miuk · 2023-10-16

**Soundness:** 2 fair
**Presentation:** 2 fair
**Contribution:** 2 fair
**Rating:** 6
**Confidence:** 4

**Summary:**

This paper proposes a neural subset selection method based on deep sets. This model is inspired by a theoretical perspective to include information from supersets to achieve better performance. Experiments on common benchmarks show SOTA performance compared to several recent baselines.

**Strengths:**

1. The idea to include information from superset is simple and effective as shown by the experiment results
2. Theoretical discussions are provided.

**Weaknesses:**

1. Equation 4 describes the neural network construction. However, I am unclear about the objective function to optimize the neural network. Also, after optimization, how do you use this neural network to select a subset?

2. In equation 4, how do you divide a superset into several subsets? There are an exponential number of combinations.

3. What is the number of learnable parameters for each baseline method and the proposed method?

**Questions:**

None

---

> ### Author Response · Authors · 2023-11-14
> **Answers to Reviewer miuk (part 1)**
>
> We greatly appreciate the time and effort you have invested. In response to your concerns, we have provided detailed clarifications.
> ### ***Comments 1: The optimization objective and how to select an optimal subset during inference?***
> ***ANSWER:*** Thank you for your insightful question, as the construction of an optimization objective and inference process are indeed key aspects of neural subset selection [1, 2, 3]. We aimed to balance detailed explanation with readability for a broad audience, which is why we initially provided a high-level overview in the **Introduction Section**. To specifically address your concerns, we are now including more detailed descriptions of the optimization and inference processes.
>
> Our formulation of the optimization objective is based on the framework established in [1]. Specifically, the optimization objective is to address Equation 1 in our paper by adopting an implicit learning strategy grounded in probabilistic reasoning. This approach can be succinctly formulated as follows:
> $$ argmax_\theta\ \mathbb{E}_{\mathbb{P}(V, S)} [\log p _\theta (S^{*}| V)] $$
> $$s.t.  p _\theta (S | V) \propto  F _\theta (S ; V), \forall  S \in 2^V, $$
>
> The important step in addressing this problem involves constructing an appropriate set mass function $p_\theta (S|V)$ that is monotonically increasing in relation to the utility function $F_\theta (S;V)$. To achieve this, we can employ the Energy-Based Model (EBM):
> $$
> p_\theta (S|V) = \frac{\mathrm{exp}( F_\theta (S; V))}{Z}, \; Z := \sum\nolimits_{S'\subseteq V}  \mathrm{exp}( F_\theta (S'; V)),
> $$
> In practice, we approximate the EBM by solving a variational approximation
> $$
>     \phi^* =  argmin_{\phi} D(q_\phi(Y|S,V)) || p_\theta (S|V)),
> $$
> During the training phase, we need an EquiNet, denoted as $Y = \operatorname{EquiNet}(V;\phi): 2^V \rightarrow [0,1]^{|V|}.$ This network takes the ground set $V$ as input and outputs probabilities indicating the likelihood of each element $x \in V$ being part of the optimal subset $S^*$. In the inference stage, EquiNet is employed to predict the optimal subset for a given ground set $V$, using a TopN rounding approach. For detailed information on the implementation and derivation of the aforementioned objective, please refer to [1].
>
> [1] Ou Z, Xu T, Su Q, et al. "Learning Set Functions Under the Optimal Subset Oracle via Equivariant Variational Inference."NeurIPS, 2022.
>
> [2] Tschiatschek S, Sahin A, Krause A. "Differentiable submodular maximization." IJCAI, 2018.
>
> [3] Zhang D W, Burghouts G J, Snoek C G M. "Set prediction without imposing structure as conditional density estimation." ICLR, 2021.

---

> ### Author Response · Authors · 2023-11-14
> **Answers to Reviewer miuk (part 2)**
>
> ### ***Comments 2: In equation 4, how do you divide a superset into several subsets?***
> ***ANSWER:*** Theorem 3.5 and Eq.4 are general frameworks to establish the relationship between $Y$ and $(S,V)$. This approach **does not necessitate** dividing a superset into multiple subsets; instead, it requires processing only once for a specific pair of $(S,V).$ In the context of neural subset selection with the optimal supervision (OS) oracle, addressing the variational approximation employs Monte-Carlo (MC) sampling. For a given $V_i,$ we only generate $m$ subsets during training, consistently setting this number to $5$ across various tasks. Consequently, this **eliminates the need for an exponential number of combinations**. It is important to highlight that one of the foremost advantages of Neural Subset Selection in the context of OS Oracle is its ability to **significantly reduce the computational burden** associated with processing an exponential number of $(S,V).$
>
> ### ***Comments 3: What is the number of learnable parameters for baselines and the proposed method?***
> ***ANSWER:*** In regard to the parameters, we have already done ablation studies in **Table 3** and provided a discussion in **Section 4.4** of our paper. To further demonstrate that the improvements achieved by our method are **not merely due to additional parameters**, we present an additional table  here using the CeleA dataset. This table compares EquiVSet (v1) and EquiVSet (v2) — variants of EquiVSet where we have incorporated a Conv(32, 3, 2) layer and a Conv(64, 4, 2) layer into the EquiVSet backbone, respectively. Detailed descriptions of these backbones are available in Appendix E.2. Notably, despite having the largest number of parameters, EquiVSet (v2) is outperformed by INSET, indicating that **our method's efficacy is not solely parameter-dependent**.
>
> |  | DeepSet | Set-Transformer | EquiVSet | EquiVSet-v1  | EquiVSet-v2 | INSET
> |--|--|--|--|--|--|--|
> | Parameter |  651181 | 1288686 | 1782680 | 2045080 | **3421592** | 2162181
> |MJC| 0.440$\pm$0.006| 0.527$\pm$0.008 | 0.549$\pm$0.005 | 0.554$\pm$0.007 | 0.560$\pm$0.005 | **0.580$\pm$0.012**
>
> Thank you in advance for dedicating your time and attention to our response. We are confident that the clarifications and additional information provided here comprehensively address your concerns. With this in mind, we respectfully and earnestly request that you re-evaluate our work, considering the explanations we have offered.

---

> > ### Comment · Reviewer_miuk · 2023-11-14
> > **Need more clarification on Comments 2**
> >
> > Thanks for your response! I have one more question regarding Comments 2. By stating "we only generate m
> >  subsets during training", do you mean that during each training iteration, you randomly select m subsets?

---

> ### Author Response · Authors · 2023-11-15
> **Thank you for your prompt reply**
>
> Thank you for your prompt response! You are correct that during each training iteration, we randomly select $m$ subsets for each ground set $V$. Increasing the value of $m$ leads to an improvement in performance. To ensure a **fair comparison**, we adhere to EquiVSet's protocol by setting the sample number $m$ to 5 **across all tasks and datasets**. Even when varying the value of $m$, the results **consistently** demonstrate that INSET **significantly outperforms** EquiVSet. In the following table, we report the performance of EquiVSet by selecting the best results achieved after tuning the value of $m$ within the range of 1 to 10.
>
> | | EquiVSet | m=1 | m=2 |m=5 | m=7 |m=8 | m=10
> |--|--|--|--|--|--|--|--|
> | Toys |  70.4$\pm$0.004 | 75.2$\pm$0.006 | 75.3$\pm$0.005 | 76.9$\pm$0.005 | 76.8$\pm$0.003 |76.7$\pm$0.003| **77.1$\pm$0.004**
> | Gear| 74.5$\pm$0.013 |78.8$\pm$0.015|  77.5$\pm$0.020| 80.8$\pm$0.012 | 81.3$\pm$0.010 | 82.1$\pm$0.015 | **84.6$\pm$0.011**
>
> Thank you for your time. If you have any additional questions, we would be delighted to discuss them further.

---

> > ### Comment · Reviewer_miuk · 2023-11-15
> > **Response**
> >
> > I am satisfied with the clarification and increased my score to 6.

---

> ### Author Response · Authors · 2023-11-15
> **Thanks for raising the score!**
>
> Huge thanks for your super quick reply and for raising the score!

---

> ### Author Response · Authors · 2023-11-23
> **Revised Manuscript Incorporating Your Suggestions.**
>
> Dear Reviewer miuk,
>
> We would like to extend our heartfelt gratitude for your active engagement and valuable suggestions. Thanks to your insightful feedback, we have made some revisions to our manuscript.
>
> Firstly, we have incorporated the optimization objective and inference process into Appendix D.2, allowing for a more comprehensive understanding of our proposed approach. Additionally, we have included the extra experiments in Appendices F.2 and F.4, providing further supporting evidence for our findings. These revisions have been highlighted in purple for readers' convenience.
>
> We greatly appreciate your continued support and acknowledgement of our response. Moreover, we are truly grateful for the time and consideration you have invested in reviewing our manuscript.
>
> Sincerely,
>
> The Authors

---

### Official Review · Reviewer_vcJR · 2023-11-11

**Soundness:** 3 good
**Presentation:** 3 good
**Contribution:** 3 good
**Rating:** 5
**Confidence:** 3

**Summary:**

The paper tackles neural subset selection. In particular, they tackle the issue that current methods do not consider the properties of the superset while constructing subsets. Their theoretical findings demonstrate that when the target value is conditioned on both the input set and subset, it is essential to incorporate an invariant sufficient statistic of the superset into the subset of interest for effective learning.

**Strengths:**

- The paper is clearly written.
- The related work covers enough ground for a new researcher to understand a high level idea of this field.
- The experiments include multiple baselines.

**Weaknesses:**

- Lack of ablation studies.
- The proposed method is not evaluated on a wide distribution of datasets.
- Will similar findings hold if the dataset contains imbalance? If so, what degree of imbalance do the guarantees still hold?

**Questions:**

- Baselines do not consider the information from superset, but these baselines be improved by adding the invariant sufficient statistic of the superset?

---

> ### Author Response · Authors · 2023-11-14
> **Answers to Reviewer vcJR (part 1)**
>
> We greatly appreciate the time and effort you have invested. In response to your concerns, we have provided detailed clarifications and additional experimental results. For your convenience, these results are presented in tabular format. We will incorporate these results, along with details and plots into the appendix of the revised version.
>
> ### ***Comments 1: Lack of ablation studies***
>
> ***ANSWER***: Thank you for highlighting the absence of ablation studies in our work. Indeed, our method, INSET, **does not introduce any new hyperparameters** to the EquiVSet [1] framework. We use the **exact same** hyperparameters as EquiVSet in all of our experiments, ensuring a **fair comparison**. Meanwhile, INSET can significantly outperform baseline models across various datasets and tasks, demonstrating its **substantial efficacy**.
>
> To further verify the robustness of INSET, we have now conducted ablation studies focusing on the Monte-Carlo (MC) sample numbers for each input pair $\{(V_i, S_i^*)\}$. In the context of neural subset selection tasks, our primary aim is to train the model  $\theta$ to predict the optimal subset $S^*$  from a given ground set $V$.  During training, we sample m subsets from $V$ to optimize our model parameters  $θ$, thereby maximizing the conditional probability distribution $p_\theta (S^* | V)$ among of all pairs of $(S,V)$ for for a given V. In our main experiments, we adhere to EquiVSet's protocol by setting the sample number $m$ to 5 across all the tasks. Even with varying the value of $m$, the results **consistently** demonstrate that INSET **significantly outperforms** EquiVSet. Please note that the performance of EquiVSet is reported by **selecting the best results** achieved after tuning the value of $m$.
>
> | | EquiVSet | m=1 | m=2 |m=5 | m=7 |m=8 | m=10
> |--|--|--|--|--|--|--|--|
> | Toys |  0.704$\pm$0.004 | 0.752$\pm$0.006 | 0.753$\pm$0.005 | 0.769$\pm$0.005 | 0.768$\pm$0.003 | 0.767$\pm$0.003| **0.771$\pm$0.004**
> | Gear| 0.745$\pm$0.013 | 0.788$\pm$0.015|  0.775$\pm$0.020| 0.808$\pm$0.012 | 0.813$\pm$0.010 | 0.821$\pm$0.015 | **0.846$\pm$0.011**
> | Bath| 0.820$\pm$0.005 | 0.821$\pm$0.010|  0.851$\pm$0.008| 0.862$\pm$0.005| 0.874$\pm$0.006 | 0.861$\pm$0.005 | **0.874$\pm$0.003**
> | Health| 72.0$\pm$0.010 | 0.749$\pm$0.015|  0.763$\pm$0.012| 0.812$\pm$0.005 | **0.824$\pm$0.008** | 0.808$\pm$0.005 | 0.811$\pm$0.005
>
> ###  ***Comments 2: The proposed method is not evaluated on a wide distribution of datasets.***
>
> ***ANSWER***: It is important to clarify that our experiments encompass three tasks: product recommendation, set anomaly detection, and compound selection, which involve the processing of tabular data, images, and 3D Cartesian coordinates. Specifically, we conduct extensive experiments on **these tasks using six datasets**. Notably, for the product recommendation task, the datasets consist of 12 categories, effectively representing **12 sub-datasets**.
>
> Moreover, we have also conducted synthetic experiments in Appendix F.1 to assess INSET's capability to learn complex set functions.  Additionally, to provide further evidence of INSET's effectiveness, we have performed set anomaly detection tasks using the CIFAR-10 dataset. We are also incorporating additional filters for compound selection tasks for a wider distribution of datasets. For more information, please refer to Appendix F.3 and F.5 in the revised submission."
>
> |  | Random | PGM | DeepSet | Set-Transformer | EquiVSet | INSET
> | :-: | :-: | :-: | :-: | :-: | :-: | :-: |
> | CIFAR-10 | 0.193 |  0.450$\pm$0.020 | 0.316$\pm$0.008 | 0.654$\pm$0.023 | 0.603$\pm$0.012 | **0.742$\pm$0.020**
> | PDBBind | 0.073 | 0.350$\pm$0.009 | 0.323$\pm$ 0.004 | 0.355$\pm$0.010 | 0.357$\pm$0.005 | **0.371$\pm$0.010**
> | BindingDB | 0.027 | 0.176$\pm$0.006 | 0.165$\pm$0.005 | 0.183$\pm$0.004 | 0.188$\pm$0.006 | **0.198$\pm$0.005**
>
> The latest results provide further evidence of INSET's superior performance compared to the baselines. Furthermore, it is worth mentioning that our experimental setup includes a **significantly larger number** of experiments compared to DeepSet (Sec. 4.3) [2] and PGM (Sec. 5.3) [3].
>
> [1] Ou Z, Xu T, Su Q, et al. "Learning Set Functions Under the Optimal Subset Oracle via Equivariant Variational Inference."NeurIPS, 2022.
>
> [2] Zaheer M, Kottur S, Ravanbakhsh S, et al. "Deep Sets." NeurIPS, 2017.
>
> [3] Tschiatschek S, Sahin A, Krause A. "Differentiable submodular maximization." IJCAI, 2018.

---

> ### Author Response · Authors · 2023-11-14
> **Answers to Reviewer vcJR (part 2)**
>
> ###  ***Comments 3: Will similar findings hold if the dataset contains imbalance? What degree of imbalance do the guarantees still hold?***
>
> ***ANSWER***: INSET is designed to significantly enhance the models' capacity to effectively learn $P(Y|S,V)$ or $F(S,V)$. According to Theorem 3.5, this enhancement **holds true consistently** when the tasks involve modeling the relationship between (S,V) and $Y$. To provide empirical evidence, we conduct additional experiments that demonstrate INSET's consistent superiority over the baselines, even in scenarios with imbalanced ground set sizes. Specifically, we train the model on the two-moons datasets (for detailed information, please refer to Appendix F.1) using fixed ground set sizes of 100, and evaluate its performance on various data sizes ranging from 200 to 1000.
>
> |  | 200  | 400 | 600 | 800| 1000 |
> |--|--|--|--|--|--|
> |EquiVSet | 0.538 $\pm$ 0.002  | 0.513 $\pm$ 0.003 | 0.482 $\pm$ 0.002 | 0.473  $\pm$ 0.005 | 0.471 $\pm$ 0.003
> |INSET |  0.547 $\pm$ 0.002 | 0.518 $\pm$ 0.005 | 0.502 $\pm$ 0.003 | 0.486 $\pm$ 0.002 | 0.485 $\pm$ 0.002
>
> The results clearly show that INSET **consistently enhances** the performance of EquiVSet, regardless of any imbalances.
>
>
> ### ***Comments 4: Can baselines be improved by adding the invariant sufficient statistic of the superset?***
>
> ***ANSWER:*** Certainly, Theorem 3.5 offers a comprehensive framework for modeling the relationship between $Y$ and $(S,V)$, which is also **applicable to the baselines**. However, integrating this invariant sufficient statistic directly into DeepSet and Set-Transformer presents challenges, as they do not explicitly learn a neural subset function $F(S,V)$. Our method, INSET, has employed DeepSet as its backbone. To demonstrate that Set-Transformer can also derive benefits from INSET, we utilize Set-Transformer as our backbone to showcase this.
>
> |  | Random  |  Set-Transformer | Set-Transformer + INSET
> |--|--|--|--|
> | Toys |0.083 | 0.625 $\pm$ 0.020 | **0.769 $\pm$ 0.010**|
> | Gear | 0.077 | 0.647 $\pm$ 0.006| **0.825 $\pm$ 0.021** |
> | Carseats | 0.066 | 0.220 $\pm$ 0.010| **0.230 $\pm$ 0.031**|
> | Bath | 0.076 | 0.716 $\pm$ 0.005| **0.862 $\pm$ 0.005**|
> | Health |0.076  |0.690 $\pm$ 0.010 | **0.852 $\pm$ 0.009** |
> | Diaper | 0.084 |0.789 $\pm$ 0.005 | **0.896 $\pm$ 0.005** |
> | Bedding | 0.079 | 0.760 $\pm$ 0.020| **0.885 $\pm$ 0.013** |
> | Feeding | 0.093 | 0.753 $\pm$ 0.006| **0.902 $\pm$ 0.004** |
>
> By employing Set-Transformer as our backbone, we enable it to explicitly learn the relationship between $Y$ and $(S,V)$. The empirical results clearly demonstrate a **significant improvement** in performance as a result.
>
> Thank you for your time and thoughtful consideration. If you have any concerns or questions, please don't hesitate to reach out to us.

---

> ### Author Response · Authors · 2023-11-16
> **Summary of our response**
>
> Dear Reviewer vcJR:
>
> Thank you again for your valuable time and efforts in reviewing our manuscript. Since our previous response was a bit long, we provide a summary below:
>
> - **Ablation studies:** We have clarified that our method INSET does not introduce new hyper-parameters. Additionally, we have included an ablation study focusing on the number of Monte-Carlo (MC) samples. Detailed responses to these points are available in our feedback to Comment 1.
>
> - **Datasets:** We have elaborated on the usage of our datasets, encompassing 3 tasks across 6 datasets in three different modalities. Besides, we have also presented more experiments in our feedback to Comment 2.
>
> - **Questions:** We also provide new experiments to answer your thoughtful questions on the imbalance and baseline.
>
> Our method not only demonstrates an impressive empirical performance, with up to a 23% improvement over the best baselines, but it is also underpinned by rigorous theoretical analysis and a strong foundational concept. We are also grateful for your recognition of our work’s Soundness, Presentation, and Contribution as being satisfactory.
>
> Considering these aspects, we respectfully and kindly invite you to re-evaluate the rating of our submission. We eagerly anticipate any further feedback from you.

---

> ### Author Response · Authors · 2023-11-19
> **Update Appendix in Response to Your Concerns**
>
> Dear Reviewer vcJR,
>
> Thanks for your time and consideration. We have revised the Appendix of our manuscript to address your concerns. Regarding your comments on ablation studies, we have conducted additional experiments, detailed in Appendix F.4, and in the first answer of our initial [response](https://openreview.net/forum?id=eepoE7iLpL&noteId=29tZ7OiyiO). Concerning the distribution of datasets, we invite you to review Appendix F.3 and F.5, along with the second answer in our initial [response](https://openreview.net/forum?id=eepoE7iLpL&noteId=29tZ7OiyiO). For your insightful questions, please refer to the second part of our initial [response](https://openreview.net/forum?id=eepoE7iLpL&noteId=2kWyNTwmG6). A summary of our previous response can be found in the [paragraph](https://openreview.net/forum?id=eepoE7iLpL&noteId=jZucIfWVeG). We would appreciate knowing if you have any additional feedback or suggestions.
>
> Sincerely,
>
> The Authors

---

> ### Author Response · Authors · 2023-11-22
> **Gentle Reminder of the Revision Deadline**
>
> Dear Reviewer vcJR,
>
> Thank you once again for your time! We understand that you have a busy schedule, and we kindly remind you that the revision deadline is approaching. If you have any suggestions or feedback on how we can improve our manuscript, we would greatly appreciate your input. We eagerly await your response.
>
> Sincerely,
>
> The Authors

---

> ### Author Response · Authors · 2023-11-26
> **Look forward to your feedback!**
>
> Dear Reviewer vcJR,
>
> We sincerely appreciate the time and effort you have dedicated to reviewing our work. Would you mind checking our response (a [shortened summary](https://openreview.net/forum?id=eepoE7iLpL&noteId=jZucIfWVeG), and the [details](https://openreview.net/forum?id=eepoE7iLpL&noteId=29tZ7OiyiO) )? If you have any further questions or concerns, we would be grateful if you could let us know. Moreover, if you find our response satisfactory, we kindly ask you to consider the possibility of improving your rating. Thank you very much for your valuable contribution.
>
> Best,
>
> The Authors

---

> ### Author Response · Authors · 2023-12-01
> **Less Than One Day Remaining for Discussion**
>
> Dear Reviewer vcJR,
>
> As the deadline for updating our manuscript is rapidly approaching, we would greatly appreciate your timely feedback on the revisions and clarifications we have provided. We are eager to incorporate any further suggestions you may have. If you find our responses and modifications satisfactory, we kindly request that you consider revising your rating to reflect these changes.
>
> Thank you for your attention to our work, and we look forward to your response.
>
> Best regards,
>
> The Authors.

---

### Official Review · Reviewer_PoVi · 2023-11-21

**Soundness:** 3 good
**Presentation:** 2 fair
**Contribution:** 3 good
**Rating:** 6
**Confidence:** 3

**Summary:**

The authors propose an optimal subset selection method based on neural networks, which is designed to learn a permutation invariant representation of both the subset of interest $S$ and the ground superset $V$. The authors highlight that prior works for neural subset selection (e.g., DeepSet) do not account for the superset $V$, and both theoretically and empirically demonstrate that jointly modeling the interactions between $S$ and $V$ leads to improved performance.

**Strengths:**

- The writing is generally easy to follow, and the paper includes a sufficiently comprehensive discussion of relevant prior works. Experimental results are presented well.
- The proposed method achieves strong empirical performance in terms of mean Jaccard coefficient (often with a fairly large gap) when compared against several optimal subset selection baselines (e.g., DeepSet, EquiVSet).

**Weaknesses:**

- The presentation of some of the mathematical details needs improvement. In particular, it seems that some of the notations are overloaded (i.e., the same notation is used with different interpretations) or not clearly defined. For example, the notation $S$ appears as a *subset* of the ground set $V$ in the Introduction, but in Section 3.1 (Background), the notation $S$ appears as an *element* of $V$ that takes a matrix form. The relationship between elements $x_i \in \mathcal{X}$ and $S_i$ is not clearly defined either. On another note, it is not entirely clear to me what the function value $Y \in \mathcal{Y}$ is really referring to, which also appears without an explicit discussion of its meaning in the Introduction as part of the variational distribution $q(Y|S,V)$. Is $Y \in \mathcal{Y}$ supposed to be the utility function value (which was also introduced with the notation $U = F_{\theta}(S,V)$ in the Introduction)? The confusion arising from notational ambiguity makes the paper less readable.

**Questions:**

- Can the authors clearly define what $Y$ is? The footnote mentions that $Y_i$ is the "probability of element $i$ being selected", but this description is ambiguous.
- It looks like learning the neural network approximation in Eq. (4) is done via variational inference as in Ou et al. (2022). As I am not familiar with the cited work, it is unclear to me how $q(Y|S,V)$ is serves as an approximation for the subset likelihood $p(S|V)$ when the former is a distribution over $Y$ and the latter is a distribution over $S$. Can the authors provide clarifications on this?
- How is the neural network construction in Eq. (4) explicitly related to $p_{\theta}(S,V)$ (or $F_{\theta}(S,V)$)?

---

> ### Author Response · Authors · 2023-11-21
> **Answers to Reviewer PoVi**
>
> We greatly appreciate the time and effort you have invested! In response to your concerns, we have provided clarifications here. We will also incorporate these clarifications into our revised version to enhance clarity.
>
> ### ***Comments 1: Relationship Between $x_i, S_i,$ and $V.$***
> ***ANSWER:*** We regard $V$ as a set composed of $n$ elements, denoted as $x_i$, i.e., $V=\\\{x_1, x_2,..., x_n\\\}$. In order to facilitate the proposition of Property 1, we describe $V$ as a collection of several disjoint subsets, specifically $V = \\\{S_1, \dots, S_m\\\}$, where $S_i \in \mathbb{R}^{n_i \times d}$. Here, $n_i$ represents the size of subset $S_i \subset V$, that is, $S_i = \\\{x_{1_i}, x_{2_i},..., x_{n_i}\\\}.$
>
> ### ***Comments 2: The definition of $Y.$***
> ***ANSWER:*** The generality of our Theorem 3.1 allows it to be applied to both U=F(Y|S,V) and P(Y|S,V) for different tasks. Specifically, when considering the task of Neural Subset Selection in Optimal Subset (OS) oracles, which involves learning P(Y|S,V), we define Y as a $|V|$ independent Bernoulli distribution, which is parameterized by $Y \in [0,1]^{|V|}$, representing the odds or probabilities of selecting element $x_i \in V$ in a output subset $S$.
>
> ### ***Comments 3: Why can $q(Y|S,V)$ serve as a variational approximation to $P(S|V)?$***
> ***ANSWER:*** As discussed in the previous answer, $Y \in [0,1]^{|V|}$. In practice, $S$ is represented as a binary vector (mask), denoted as $S := \\\{0,1\\\}^{|V|}$, where the $i$-th element is equal to $1$ if $i \in S$ and $0$ otherwise. Therefore, it is natural to use $q(Y|S,V)$ to represent the variational distribution of $P(S|V)$.
>
> ### ***Comments 4: How is the neural network construction in Eq. (4) explicitly related to $p_\theta(S,V)$ or $F_\theta(S,V).$***
> ***ANSWER:*** Once neural networks are trained, their outputs become fixed for a given input, such as (S,V). Thus, Eq. (4) represents the explicit structure used to construct models for learning the deterministic function $\theta(S,V)$ (to differentiate it from the utility function U=F(S,V)). Using this function, we can construct the conditional distribution $q(Y|S,V)$ according to Theorem 3.5. Specifically, we employ the Mean-Field Variational Inference (MFVI) method introduced by [1] (Section 3.2) to approximate the distribution $q(Y|S,V)$, referred to as $\psi$ in [1].
>
> To prevent overwhelming readers with an abundance of notations and equations, we have deliberately omitted the detailed construction of q(Y|S,V) and the derivation of variational approximation in our paper. This decision was motivated by two factors. Firstly, our theorem and Eq. 4 offer a general framework for modeling the relationship between $Y$ and $(S,V)$, instead of focusing on the neural subset selection tasks. Secondly, in order to ensure clarity of our motivation, we have provided a high-level description of these concepts in the Introduction section. For readers interested in the details of these concepts, we strongly recommend referring to [1] (Section 3) for a more comprehensive understanding. For the implementation details of q(Y|S,V) and $\theta(S,V)$, we suggest consulting our accompanying code located at (./model/modules.py).
>
> Thanks for your time and suggestions again. We would appreciate knowing if you have any additional feedback or suggestions.
>
> [1] Ou Z, Xu T, Su Q, et al., "Learning Set Functions Under the Optimal Subset Oracle via Equivariant Variational Inference."NeurIPS, 2022.

---

> ### Author Response · Authors · 2023-11-23
> **Revision of Manuscript Incorporating Your Suggestions**
>
> Dear Reviewer PoVi,
>
> We sincerely appreciate your reviews and valuable suggestions. Taking into account your feedback, we have made refinements to the footnote in the Introduction Section and enhanced the description of $V$ and $S$ in Section 3.1. These revisions, highlighted in purple, will significantly enhance the clarity of our paper. Thank you once again for your time and contribution.
>
> Best regards,
>
> The Authors.

---

> > ### Comment · Reviewer_PoVi · 2023-11-30
> > **Follow-up clarification questions and comments**
> >
> > Thanks for a quick response and for letting me know of the revisions. Here are additional clarification questions and comments:
> > - Regarding Comment 1: Wouldn't it be more natural to write $V = S_1 \cup \cdots \cup S_m$ with $S_i \cap S_j = \emptyset$ when $i \neq j$, which I believe is indeed the form used in Section 3.2?
> > - Regarding Comments 2 and 3: I think the description that $Y$ is a *distribution* is misleading. Clearly, $Y$ itself is not a distribution since it need not be the case that $\sum_{i} Y_i = 1$, and by the definitions given here, $P(S|V)$ and $q(Y|S,V)$ are distributions over different objects. Rather, shouldn't it be the case that the probabilities of each element being selected in the optimal subset are the outputs from the variational distribution $q$? In this case, it seems more appropriate to describe it as $q(S|V)$? Please let me know if I am misunderstanding something here. Meanwhile, since $Y$ is used throughout the main text, I think it should be very clearly defined as part of the main text before it is used, instead of appearing in a footnote (if appropriate, discussed along with concrete examples that readers can immediately understand).

---

> > > ### Author Response · Authors · 2023-12-01
> > > **Thanks for your reply and suggestions!**
> > >
> > > Thank you for your valuable suggestions! We are now revising our manuscript based on your suggestions. To address your questions, we would like to provide the following clarifications:
> > >
> > > Firstly, we want to clarify that $Y \in [0, 1]^{|V|}$ represents a set of $|V|$ **independent Bernoulli distributions** rather than a categorical distribution with $V$ classes. Hence, it is not required for the elements of $Y$ to satisfy the constraint $\sum_i Y_i = 1$. Additionally, we define the expression of $q(Y|S,V)$ as follows:
> > > $$
> > > q(Y|S,V) = \prod_{i \in S}Y_i \prod_{i \not\in S}(1-Y_i), Y\in[0,1]^{|V|}.
> > > $$
> > > Next, we would like to explain why $q(Y|S,V)$ can approximate $P(S|V)$. Consider $Y \in [0, 1]^{|V|}$ and $S = \\\{0, 1\\\}^{|V|}$. In this case, $Y$ can be viewed as a stochastic version of $S$ since $Y$ can also be generated as $Y = \\\{0, 1\\\}^{|V|}$ while still satisfying the constraint $Y \in [0, 1]^{|V|}$.
> > >
> > > To facilitate comprehension, let us consider an illustrative scenario. Suppose we have a ground set $V = \\\{x_1, x_2, x_3\\\}$, and the optimal subset $S^*$ is $\\\{x_1, x_2\\\}$, which can be represented as $[1, 1, 0]$. Specifically, we define $P(S^*|V) = 1$, indicating that $S^*$ is the correct subset, while for any $S \neq S^*$, we have $P(S|V) = 0$.
> > >
> > > Now, let's examine the case when $Y = [1, 1, 0].$ In this situation, we can calculate that $q(Y|S^,V) = 1.$ This implies that $q(Y|S^,V)$ accurately represents the probability of observing $S^*$ given $V$, and it correctly assigns a high probability to the optimal subset.
> > >
> > > We hope these clarifications help provide a better understanding of our framework. Once again, we appreciate your constructive suggestions and look forward to further discussion.

---

> > > ### Author Response · Authors · 2023-12-02
> > > **Discussion Stage 1 is Near the End: Do You Have Any Further Suggestions?**
> > >
> > > Dear Reviewer PoVi,
> > >
> > > Thank you for your constructive suggestions and insightful comments! As the discussion deadline is approaching in 10 hours, we would like to inquire if you have any further suggestions for improving our manuscript. We would greatly value your input and appreciate your guidance.
> > >
> > > Thank you for your time and consideration. We eagerly await your reply.
> > >
> > > Sincerely,
> > > The Authors

---

> ### Author Response · Authors · 2023-12-01
> **The manuscript has been updated**
>
> Dear Reviewer PoVi,
>
> We sincerely appreciate your valuable suggestions for improving our paper. We have refined the descriptions of $V$ and $S$ in Section 3.2 based on your suggestions. Moreover, we have defined $Y$ in the main text instead of the footnote, where we have also included a reference to Appendix D.2. In this appendix, readers can find further elaboration on the relationship between $q(Y|S,V)$ and $P(S,V), along with the example mentioned in our previous response.
>
> We sincerely thank you once again for your time and valuable contribution. Should you have any additional suggestions or questions, please do not hesitate to let us know.
>
> Best regards,
> The Authors

---

### Author Response · Authors · 2023-11-28
**Summary of Review and Response (December 1st, UTC-12)**

We express our sincere gratitude to the Area Chairs and Reviewers for their dedicated time and valuable feedback. Below is a concise summary of the review and our responses for ease of reference.

***Reviewer Acknowledgments***:

Our method, INSET, has been recognized for its **strong empirical performance** and **rigorous theoretical support**. Key highlights include:

- **Clear Motivation with Theoretical Support:** Our model is inspired by a **theoretical result** [miuk], and have theoretically demonstrated the significance to include information from supersets to achieve better performance [PoVi, vcJR].
- **Strong Empirical Performance:** INSET achieves **state-of-the-art** performance [miuk], supported by empirical evidence [PoVi].
- **Quality of Writing:** The manuscript is praised for its clarity and simplicity, making it **accessible for newcomers** to the field [vcJR, miuk]. The presentation **effectively conveys high-level concepts** [PoVi].

***Addressing Weaknesses***:

The reviewers raised concerns regarding ablation studies [vcJR, miuk], dataset distribution [vcJR], and mathematical notations in the optimal subset oracle [PoVi, miuk]. Our responses include:

- *Ablation Studies*: We clarified in this [dialog](https://openreview.net/forum?id=eepoE7iLpL&noteId=29tZ7OiyiO) that INSET introduces **no new hyperparameters**. Additional experiments demonstrate that even with variations in existing hyperparameters, INSET **consistently outperforms** baselines by **a large margin**. This response was **[acknowledged positively](https://openreview.net/forum?id=eepoE7iLpL&noteId=sIlnAhy7I9) by Reviewer miuk**.
- *Mathematical Notations:* Following suggestions from Reviewers PoVi and miuk, we corrected minor typos and enriched the appendix with additional mathematical background. These revisions received **[positive feedback](https://openreview.net/forum?id=eepoE7iLpL&noteId=sIlnAhy7I9) from Reviewer miuk**.
- *Dataset Distribution:* We clarified in the [dialog](https://openreview.net/forum?id=eepoE7iLpL&noteId=29tZ7OiyiO) that our approach **encompasses three tasks across six datasets in different modalities** in the main body of our work. The appendix have also included **five additional datasets**. We have utilized a **much wider variety of datasets** than those adopted by our baseline comparisons.



***Revision Overview***:

- To enhance clarity, we have made refinements to the description of $Y$ in the Introduction, as well as the definitions of $V$ and $S$ in Sections 3.1 and 3.2.
- The mathematical description of the optimization objective and inference process has been detailed in Appendix D.2.
- Appendix F.2 has been updated with additional empirical studies on computation costs. Further, ablation studies have been included in Appendix F.4, and results from a broader range of datasets are now presented in Appendix F.5."

We are thankful for the constructive feedback received, and we believe that the concerns raised by the reviewers have been thoroughly addressed in our responses and revisions.

---

### Meta-Review · Area_Chair_7Zun · 2023-12-18

**Metareview:**

This paper proposes a method for "neural subset selection" based on deep sets. The paper received three reviews with borderline scores. Far the most comprehensive review came from PoVi, who found the work to be easy to follow, appreciated the discussion of the previous literature and noted the comprehensiveness of the experiments. The authors took time to provide extensive responses to the reviewer complaints but the reviewers did not take sufficient time to acknowledge these responses. In this case, and given the satisfaction expressed by the few reviewers who did reply with the updated results, I tend to give the authors the benefit of the doubt and recommend acceptance.

**Justification For Why Not Higher Score:**

Too many weaknesses.

**Justification For Why Not Lower Score:**

Reviewers all see strengths.

---

### Decision · Program_Chairs · 2024-01-16

Accept (poster)